# MolDiscovery: learning mass spectrometry fragmentation of small molecules

Liu Cao [1,4], Mustafa Guler [1,4], Azat Tagirdzhanov [2,3], Yi-Yuan Lee [1], Alexey Gurevich[2] & Hosein Mohimani [1✉]

Identification of small molecules is a critical task in various areas of life science. Recent advances in mass spectrometry have enabled the collection of tandem mass spectra of small molecules from hundreds of thousands of environments. To identify which molecules are present in a sample, one can search mass spectra collected from the sample against millions of molecular structures in small molecule databases. The existing approaches are based on chemistry domain knowledge, and they fail to explain many of the peaks in mass spectra of small molecules. Here, we present molDiscovery, a mass spectral database search method that improves both efficiency and accuracy of small molecule identification by learning a probabilistic model to match small molecules with their mass spectra. A search of over 8 million spectra from the Global Natural Product Social molecular networking infrastructure shows that molDiscovery correctly identify six times more unique small molecules than previous methods.

[1] Carnegie Mellon University, Pittsburgh, PA, USA. [2] St. Petersburg State University, St. Petersburg, Russia. [3] St. Petersburg Electrotechnical University LETI, St. Petersburg, Russia. [4] These authors contributed equally: Liu Cao, Mustafa Guler. ✉email: hoseinm@andrew.cmu.edu

A crucial problem in various areas of life science is to determine which known small molecules are present/absent in a specific sample. For example, physicians are devoted to discovering small molecule biomarkers in plasma/oral/urinal/fecal/tissue samples from a patient for disease diagnosis[1] and prognosis[2]. Epidemiologists are interested in identifying small molecule disease risk factors from diet[3] and environment[4]. Ecologists are interested in characterizing the molecules produced by microbes in various microbial communities[5]. Natural product scientists need to identify all the known molecules in their sample, clearing the path towards the discovery of novel antimicrobial or antitumor molecules[6,7].

Recent advances in high-throughput mass spectrometry have enabled collection of billions of mass spectra from hundreds of thousands of host-oriented/environmental samples[8–11]. A mass spectrum is the fingerprint of a small molecule, which can be represented by a set of mass peaks (Fig. 1A, B). In order to identify small molecules in a sample with tens of thousands of spectra, one can either (i) de novo predict small molecule structure corresponding to mass spectra, (ii) search these mass spectra against tens of thousands of reference spectra in spectral libraries, or (iii) search these mass spectra against millions of molecular structures in small molecule databases. De novo prediction can potentially identify both known and novel small molecules. However, it is rarely used in practice due to the intrinsic complexity of small molecule structure and the low signal-to-noise ratio of mass spectral data. Spectral library search is recognized as the most reliable mass spectral annotation method. Nevertheless, current reference spectral libraries are limited to tens of thousands of molecules, and the majority of known small molecules are not represented in any reference spectral library. Furthermore, collecting mass spectra of all known small molecules individually would be expensive and time-consuming. The most frequently used strategy for small molecule identification is in silico search of small molecule structure databases. This approach enables small molecule identification in known databases, such as PubChem[12] and dictionary of natural products (DNP)[13]. Moreover, in silico database search also applies to discovery of novel small molecules through genome mining[14].

The majority of in silico database search methods are rule-based models that incorporate domain knowledge to score small molecule-spectrum pairs. EPIC uses heuristic penalties for matching fragment ions to score molecule-spectrum pairs[15]. MAGMa+ further scores for both matched and missed peaks[16]. MassFrontier is a commercial software that utilizes a large number of fragmentation mechanism rules to predict fragmentation spectra[17]. MetFrag is based on weighted peak count and bond dissociation energy to compute matching scores[18]. MIDAS takes account of fragment-peak matching errors[19]. MSFinder introduces the hydrogen rearrangement rules for fragmentation and scoring[20]. Unlike the other methods, QCEIMS uses quantum chemical simulation to predict mass spectra[21]. However, due to the limitation of the rules and heuristic parameters, these methods often fail to explain many of the peaks in mass spectra.

Recently, spectral libraries with tens of thousands of annotated mass spectra of small molecules have emerged, paving the path for developing machine learning based methods to improve sensitivity and specificity of in silico database search. CFM-ID applies stochastic Markov process to predict fragmentation spectra[22]. CSI:FingerID predicts a molecular fingerprint based on mass spectra and searches the fingerprint in a molecular database[23]. ChemDistiller combines both molecular fingerprint and fragmentation information to score the molecule spectrum matches[24]. However, the existing methods do not perform well for super small molecules (<400 Da), and are computationally insufficient for heavy small molecules (>1000 Da).

In this work, we improve the efficiency and accuracy of small molecule identification by (i) designing an efficient algorithm to generate mass spectrometry fragmentations and (ii) developing a probabilistic model to identify small molecules from their mass spectra. Our results show that molDiscovery greatly increases the accuracy of small molecule identification, while making the search an order of magnitude more efficient. After searching 8 million tandem mass spectra from the Global Natural Product Social molecular networking infrastructure (GNPS)[10], molDiscovery identified 3185 unique small molecules at 0% false discovery rate (FDR), a six times increase compared to existing methods. On a subset of the GNPS repository with known genomes, molDiscovery correctly links 19 known and three putative biosynthetic gene clusters to their molecular products.

## Results

**Outline of molDiscovery pipeline**. The molDiscovery pipeline starts by (i) constructing metabolite graphs and (ii) generating fragmentation graphs. For the latter, molDiscovery uses a new efficient algorithm for finding bridges and 2-cuts in the metabolite graphs. Afterward, molDiscovery proceeds with (iii) learning a probabilistic model for matching fragmentation graphs and mass spectra (Fig. 1a–e), (iv) scoring small molecule-spectrum pairs (Fig. 1f–k), and (v) computing FDR.

In the past we introduced Dereplicator+[25], a database search method for identification of small molecules from their mass spectra, that follows a similar series of steps to search mass spectra against chemical structures. However, Dereplicator+ uses a brute-force method for fragmentation graph construction, and naive shared-peak-count for scoring.

**Datasets**. The GNPS spectral library and MassBank of North America (MoNA) were used as training and testing data. For the GNPS spectral library, we selected molecule-spectrum pairs from the NIH Natural Products Library in GNPS (Oct 18, 2020) with unique spectrum (SPLASH values[26],please see Supplementary Fig. 1), unique molecule (first 14 characters of InChIKey) and precursor m/z that could be explained within an absolute error tolerance of 0.02 Da. If a molecule corresponded to multiple spectra, a random spectrum was selected. This left us with 4437 and 3964 spectra in positive and negative mode, respectively. With similar filtering steps as above and only keeping spectra with at least five peaks from Vaniya/Fiehn Natural Products Library obtained on Q-Exactive HF instruments (VF NPL QEHF), 6528 singly-charged and 163 doubly-charged spectra in positive mode were selected. We also selected 2382 singly-charged NIST20 spectra from Q-TOF instruments using the same filtering steps.

The Pacific Northwest National Laboratory (PNNL) lipid library, included as part of GNPS, was used for lipid identification benchmarking. We selected positive mode spectra with [M+H]+ adducts for which the provided lipid names in the PNNL lipid library could be resolved by the LIPID MAPS Structure Database (LMSD) text search tool. This left us with 15,917 spectra corresponding to 316 unique compounds.

We also selected multiple small molecule databases for different benchmarking tasks. DNP contained 77,057 non-redundant compounds from the dictionary of natural products[13]. MoNA DB contained 10,124 compounds from MoNA. AllDB contained 719,958 compounds from DNP, UNPD[27], HMDB[28], LMSD[29], FooDB[30], NPAtlas[31], KEGG[32], DrugBank[33], StreptomeDB[34], the GNPS spectral library molecules, MIBiG[35], PhenolDB[36], Quorumpeps[37], and in-house databases. The KNApSAcK[38] database contained 49,584 compounds of plant origin. Bioactive-PubChem (~1.3 million compounds) contained all the compounds from the PubChem

## molDiscovery training

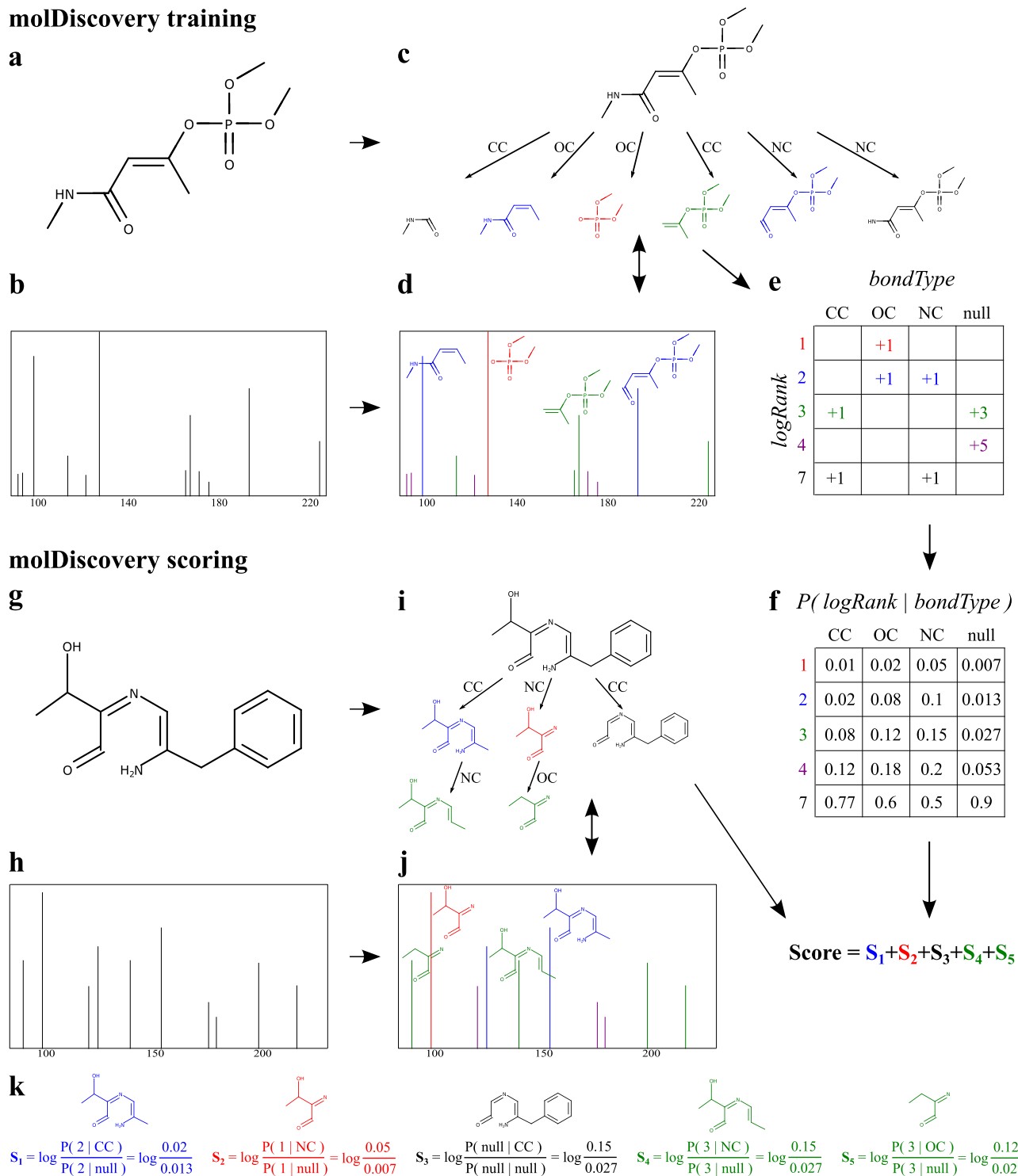

## molDiscovery scoring

$$S_1 = \log \frac{P(2 \mid CC)}{P(2 \mid null)} = \log \frac{0.02}{0.013}$$

$$S_2 = \log \frac{P(1 \mid NC)}{P(1 \mid null)} = \log \frac{0.05}{0.007}$$

$$S_3 = \log \frac{P(null \mid CC)}{P(null \mid null)} = \log \frac{0.15}{0.027}$$

$$S_4 = \log \frac{P(3 \mid NC)}{P(3 \mid null)} = \log \frac{0.15}{0.027}$$

$$S_5 = \log \frac{P(3 \mid OC)}{P(3 \mid null)} = \log \frac{0.12}{0.027}$$

database that are less than 2000 Da and have at least one reported bioactivity.

For mass spectra datasets, we selected 46 high-resolution GNPS spectral datasets (about 8 million spectra in total) with paired genomic/metagenomic data available[39] for large-scale spectral searching and secondary metabolites identification. Moreover, we selected spectral datasets from various environment, including MSV000084092 (~57,000 spectra from human serum), MSV000086427 (~209,000 spectra from 38 plant species) and MSV000079450 (~400,000 spectra from *Pseudomonas* isolates).

**Preferential fragmentation patterns in mass spectrometry.** MolDiscovery learned a probabilistic model (see "Methods") that reveals the preferences in mass spectrometry fragmentation. First, mass spectrometry fragmentation has preference for bond type *bondType*. Fig. 2a shows that the breakage of *bondType*s NC and OC leads to fragments with a higher rank (i.e., *logRank*s closer to 1) than CC bond, indicating that they are more likely to be broken by mass spectrometry. In addition, bridges (NC, OC and CC) tend to generate fragments with higher *logRank* than 2-cuts (for example, NC_NC, NC_OC, CC_OC), suggesting bridges are more likely to be fragmented than 2-cuts. Note that the *logRank*

**Fig. 1 MolDiscovery framework. a–f** These show the training procedures of molDiscovery, while **g–k** are the scoring procedures based on the pretrained probabilistic model. **a** A reference molecule $R$ in spectral library. **b** The reference spectrum of $R$. **c** The fragmentation graph of $R$. The root node represents the whole molecule, while the other nodes are fragments of it. The edges are annotated with the type of bonds where the fragmentation occur. Here, a fragment will be annotated with red, blue, green or purple if it corresponds to a mass peak in reference spectrum with $logRank = 1, 2, 3, 4$ respectively (see the "Methods" section). **d** Annotation of the reference spectrum of $R$. A mass peak will be annotated with a fragment if its mass is identical to the mass of the fragment plus charge, within a tolerance. **e** A table counting the number of fragments observed in training data with specific $bondType$ and $logRank$, referred to as count matrix. In this example, since the number of peaks is small, all the present peaks have $logRank$s 1 to 4, and the absent ones are shown with the lowest possible $logRank = 7$. Note that we allow $logRank$s between 1 and 6, corresponding to peaks with ranks between 1 and 64. The $null$ column corresponds to the experimental peaks which cannot be explained by the fragmentation graph. **f** The probabilistic model $P(logRank|bondType)$, which is computed by normalizing the count matrix. **g** A molecule $Q$ in a chemical structure database. **h** A query spectrum. **i** The fragmentation graph of $Q$. **j** Annotation of the query spectrum with the fragmentation graph of $Q$. **k** Computation of the query spectrum match score, which is the sum of scores of all the fragments in the annotated fragmentation graph. Here we represent $P(logRank = i|bondType)$ by $P(i|bondType)$. For simplicity, $logRank_{pa}$, 2-cuts columns, and rows of $logRank = 5, 6$ are not shown.

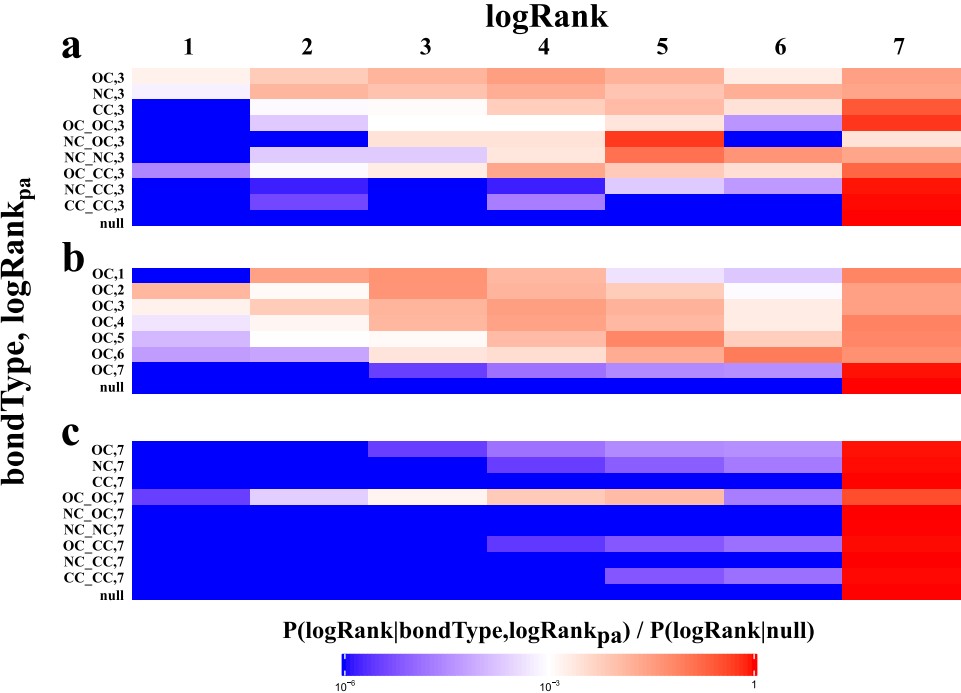

**Fig. 2 Heatmap of $P(logRank|bondType, logRank_{pa})$.** Each row represents bond type $bondType$ and parent's intensity rank $logRank_{pa}$ of a fragment. The row "null" refers to the null distribution $P(logRank|null)$. Each column represents the $logRank$ of a child fragment. When $logRank_{pa}$ is 0, it means the parent is the root (precursor molecule). **a** $logRank$ distribution over different $bondType$s for $logRank_{pa} = 3$. **b** $logRank$ distribution of OC bond over different $logRank_{pa}$. Since there is only one fragment in the fragmentation graph that could have $logRank = 1$, the parent fragment of such fragment can not generate another fragment with $logRank = 1$, hence $P(1|NC, 1) = 0$. **c** $logRank$ distribution over different $bondType$s for $logRank_{pa} = 7$. Supplementary Figs. 2 and 3 show the complete heatmaps for charge +1 and +2, respectively.

distribution of fragments due to the 2-cut CC_CC is similar to the null model distribution, defined as the distribution of log rank of peaks in the entire spectral dataset. This implies that in practice the fragmentation of CC_CC 2-cut rarely happens.

In addition, parent fragments with high ranks are more likely to produce high-rank children fragments, while fragments with low ranks are similar to random noise. For example, for fragments produced by the breakage of OC bond (Fig. 2b), if their parents have high ranks (1, 2, or 3), they are likely to be higher rank than those with lower-rank parents (6, 7). Moreover, molDiscovery automatically learns to discard fragments with low rank parents which are not informative for predicting mass spectrometry fragmentations. The distribution of $logRank$ for these fragments is similar to noise (null distribution, Fig. 2c). Therefore, the log-likelihood scores for such fragments are close to zero.

**Benchmarking the efficiency of fragmentation graph construction algorithms.** We compared the performance of the fragmentation graph construction algorithms in molDiscovery and Dereplicator+. For both algorithms, we allowed for at most two bridges and one 2-cut. The algorithms were benchmarked on DNP. Supplementary Fig. 4a shows the average fragmentation graph construction time for small molecules in different mass ranges. The results show that the average fragmentation graph construction time for Dereplicator+ grows exponentially as the mass of small molecules increase, while the construction time only grows linearly for molDiscovery. When the mass is greater than 600 Da, molDiscovery is two orders of magnitude faster than Dereplicator+.

Supplementary Fig. 4b compared the maximum memory consumption of molDiscovery with Dereplicator+. For both chemical structure databases, Dereplicator+ only works for

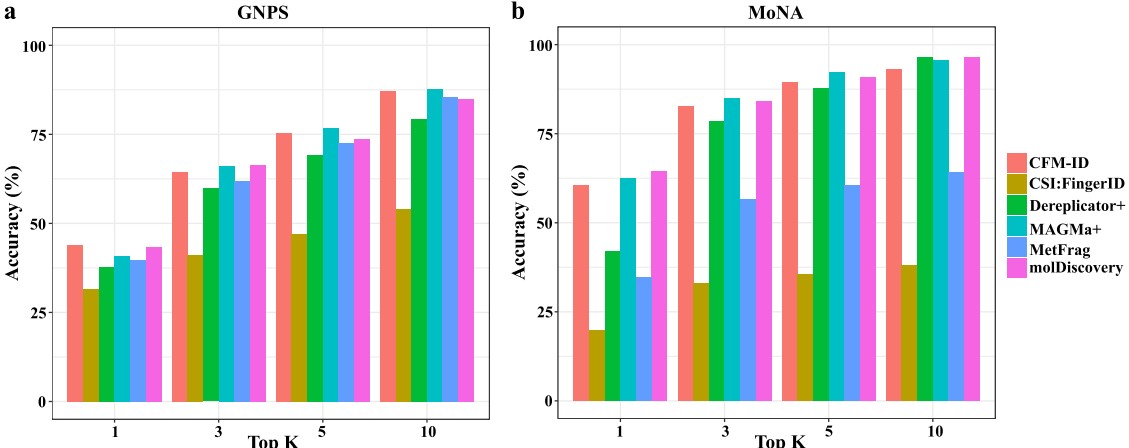

**Fig. 3 Top $K$ = 1, 3, 5, and 10 accuracy for all tested methods. a** Searching 194 spectra from the GNPS spectral library against 77,057 molecules from DNP and **b** Searching 342 spectra from MoNA against 10,124 molecules from MoNA (spectra from molecules overlapping with CSI:FingerID training data are removed). Ties in scores were evaluated by setting the rank of the candidate compounds with tied scores to the average of all of those ranks. See Supplementary Note 1 for detailed parameter settings of these methods.

---

**Table 1 Running time and peak memory usage of all methods in searching GNPS against DNP.**

| Method | Searching time (d-h:m:s) | Searching peak memory (GB) | Preprocessing time (d-h:m:s) | Preprocessing peak memory (GB) |
|---|---|---|---|---|
| molDiscovery | 00:29:04 | 0.15 | 10:40:25 | 16.38 |
| CFM-ID | 00:59:37 | 0.012 | 157-08:28:08 | 3.839 |
| Dereplicator+ | 04:28:10 | 19.98 | N/A | N/A |
| MAGMa+ | 2-20:34:37 | 0.51 | N/A | N/A |
| MetFrag | 12-08:57:24 | 4.01 | N/A | N/A |
| CSI:FingerID | 29-09:35:05 | 19.68 | N/A | N/A |

Note that CFM-ID running time and memory usage does not include filtering database candidates by precursor m/z. All running times are only for the spectra for which the method completed successfully (exit code 0). For CSI:FingerID 423 spectra either failed with a "feasibility" error or required more than 20 GB of memory, and are not included in values in this table. Note that given a small molecule structure database, molDiscovery and CFM-ID only need to preprocess the database for once. For searching very large spectral datasets, they are usually more efficient than other methods. *N/A* not applicable.
All reported running times are on a single CPU.

---

molecules with masses less than 1000 Da, while molDiscovery can handle molecules with masses up to 2000 Da.

**Benchmarking in silico database search on spectral libraries.** MolDiscovery is compared with five other state-of-the-art methods, including Dereplicator+, MAGMa+, CFM-ID, CSI: FingerID and MetFrag on the GNPS spectral library and MoNA. The database search results show that molDiscovery on average can correctly identify 43.3% and 64.3% of small molecules in the testing GNPS and MoNA data, respectively, as top-ranked identifications (Fig. 3). MolDiscovery is the best performing method in the MoNA dataset and slightly worse than CFM-ID (43.8%) in the GNPS dataset. Note that CSI:FingerID's performance significantly drops when removing its training data (Supplementary Fig. 5). Our results indicate that CSI:FingerID is biased towards the structures that are very similar to its training data, while molDiscovery's performance does not depend on the similarity between training and testing data (Supplementary Fig. 6).

MolDiscovery is also the fastest and one of the most memory-efficient methods for searching GNPS against DNP. CFM-ID is slightly slower than molDiscovery in the searching stage. In the preprocessing stage, molDiscovery is over 300 times faster than CFM-ID (Table 1).

We also evaluated the running time based on the mass ranges of the correct molecule matches (Fig. 4). We found that the highly accurate methods, CFM-ID and CSI:FingerID, have overall running times that scale poorly with molecular weight. For

spectra of molecules with masses >1000 Da we found that CSI: FingerID and CFM-ID take on average 5 h, 13 min, 10 s and 1 h, 55 min, 34 s, respectively. Meanwhile, molDiscovery takes only 6 min and 24 s on average for the same mass range.

**The effect of various bond types.** We examined a subset of NIST20 that contains S-C, O-P, and P-C bonds, and tested how accuracy is affected by addition of all the combinations of these bond types on top of our default C-C, N-C, and O-C bonds. Although adding any of these bond types increases accuracy, the largest top 1 accuracy difference is <1% compared to the default. The impact of removing C-C, N-C, and O-C bonds on the accuracy is also examined, and the results show that molDiscovery accuracy significantly drops if any of these default bond types are removed (Supplementary Fig. 7).

**Handling isobaric molecules.** Since isobaric molecules have different structures, their fragmentations generated by molDiscovery could be different (Supplementary Fig. 8). We searched the compounds from the GNPS spectral library that contain leucine or isoleucine, and created isobaric species in silico by changing leucines to isoleucines and vice versa. Among the three molecules, in two cases molDiscovery assigned a larger score to the correct molecule, while in the remaining case the scores assigned were nearly identical (Supplementary Table 1).

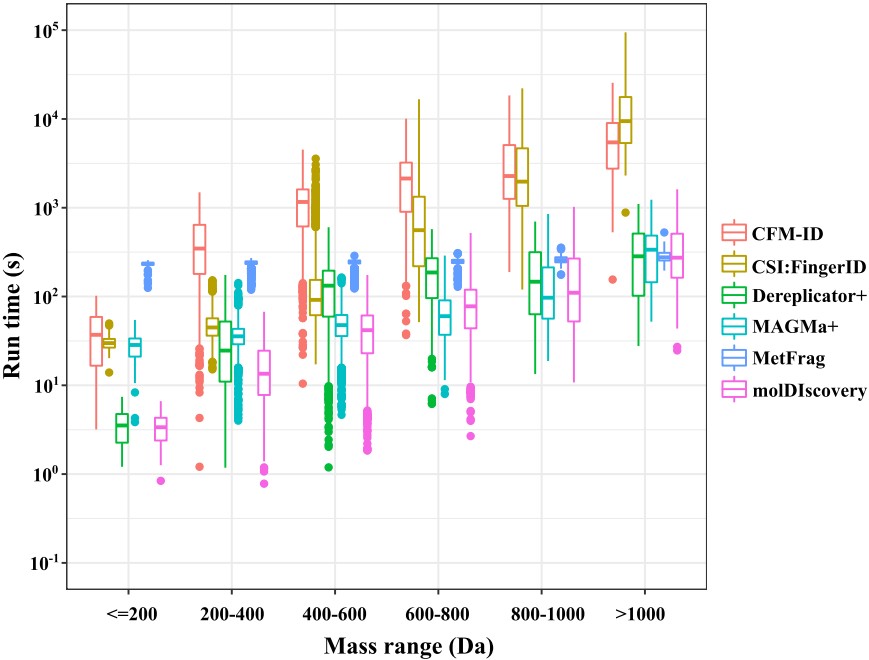

**Fig. 4 Running time comparison of different methods for compounds in the GNPS subset within different mass ranges.** Inputs that caused any method to crash or exceed 20 GB of memory usage were excluded from this analysis. All reported running times and memory usages are without preprocessing (molDiscovery, Dereplicator+ and CFM-ID could be run with or without preprocessing). In the boxplot, center lines are medians. Box limits are upper and lower quartiles. Whiskers are 1.5× interquartile range. Points are outliers.

**Evaluating molDiscovery sensitivity to platform variation**. In order to determine if the probabilistic scoring method described is robust to changes in mass spectrometry platform, a subset of annotated spectra from MoNA dataset including 1639 HCD-fragmented and 1639 CID-fragmented spectra were used for evaluation. Two models were trained using the GNPS spectra augmented with either the HCD or CID fragmented MoNA dataset, and then tested on both fragmentation modes. Searches were run against a combined chemical structure database consisting of DNP, and the corresponding MoNA and GNPS molecules. It was shown that using training data from different dissociation techniques, e.g., CID and HCD did not affect performance of molDiscovery (Supplementary Fig. 9).

**Performance on doubly-charged and negatively-charged spectra**. Only 163 spectra from MoNA dataset are annotated as doubly-charged. In light of the small amount of data we reserved the MoNA dataset as our test dataset, and used high confidence doubly-charged Dereplicator+ identifications of the GNPS MassIVE datasets as the training data (813 spectra and 180 unique molecules at cuttoff score of 15). Since this training data is still small, we bootstrapped doubly-charged parameters using the singly-charged model we previously trained. The training process updates the model parameters according to the doubly-charged training data. Despite the small training dataset, molDiscovery outperformed Dereplicator+ on search of 163 doubly-charged spectra against non-redundant DNP and GNPS chemical database augmented with MoNA molecules (Supplementary Fig. 10). Moreover, we tested molDiscovery on 3964 negatively charged spectra of the GNPS spectral library against the unique DNP database. Notably, without retraining on negative spectra, molDiscovery reached 36% accuracy (Supplementary Fig. 11).

**Performance on different compound classes and mass ranges**. To evaluate how well molDiscovery performs on different compound classes, we used ClassyFire[40] to annotate the non-redundant

GNPS spectral library. No single tool outperformed all the other tools in all the compound classes (Supplementary Fig. 12a). MolDiscovery performs better than all the other tools in two out of seven superclasses. Moreover, we compared the database search accuracies of these tools on polar and non-polar metabolites. MolDiscovery is the best performing tool for searching polar metabolites across all the polarity measures, while for non-polar metabolites there is no tool that is consistently better than others (Supplementary Fig. 14).

Similarly, molDiscovery performs equally or better than the competing methods in four out of six mass ranges (Supplementary Fig. 12b). It is worth noting that in low mass ranges (<600 Da) the prediction accuracies of all the methods are lower than their performance in high mass ranges (>600 Da), which is probably due to the fact that large molecules tend to generate more fragment ions.

**Lipid identification**. We evaluated the performance of molDiscovery for lipid identification in searching spectra from the PNNL lipid library against 39,126 unique compounds from LMSD. We found that molDiscovery achieved a top 1 accuracy of 16.9%, a top 3 accuracy of 37.4%, a top 5 accuracy of 47.6%, and a top 10 accuracy of 63.0%. Although many lipids are extremely similar to one another (Supplementary Fig. 15), notably for 173 of the 316 unique compounds in the PNNL library, molDiscovery was able to find at least one spectrum where the top-scoring match corresponded to the correct compound. This experiment was run without any re-training of molDiscovery on lipid spectra.

**Searching human and plant spectra**. To demonstrate the ability of molDiscovery to identify plant and human metabolites, we benchmarked the tool on two recent GNPS datasets. Plant spectra from 38 species (MSV000086427) was searched against AllDB complemented with the KNApSAcK database, one of the largest specialized databases of plant metabolites. Note that AllDB also includes the UNPD comprising many compounds with plant

origin[27]. Human serum spectra (MSV000084092) was searched solely against AllDB as this database has already included all compounds from the Human Metabolome Database (HMDB), the largest metabolomics database for *H. sapiens* and the LMSD, which contains biologically relevant lipids in human serum.

Since both datasets are not annotated, it is not possible to validate molDiscovery identifications by comparing them to ground truth. Instead, we compared the number of identifications in KNApSAcK/UNPD (HMDB/LMSD) to the number of identifications in the rest of AllDB for the plant (human serum) dataset. Supplementary Fig. 16 shows the number of matches from the expected database (KNApSAcK/UNPD for plant and HMDB/LMSD for human serum data) versus the number of matches among the rest of compounds in AllDB. The ratio of the identifications in the expected databases is proportionally higher than the ratio of size of the expected databases to the size of AllDB. This bias grows with the increasing cutoff score. Note that AllDB (719,958 compounds) is an order of magnitude larger than KNApSAcK (49,584), HMDB (41,919), and LMSD (40,358), and three times larger than UNPD (229,358). The bias towards compounds in the searches of plant (human serum) data indirectly demonstrates the correctness of molDiscovery results. Moreover, many hits in AllDB may also represent true positive matches since this combined database includes plant- and human-related metabolites from databases other than KNApSAcK, UNPD, HMDB, and LMSD.

**Searching large-scale spectral datasets**. We benchmarked the performance of molDiscovery, Dereplicator+ and spectral library search on 46 GNPS datasets containing a total of 8 million tandem mass spectra (Supplementary Table 2). In contrast to the GNPS spectral library where all the spectra are annotated and their molecules are known, the spectra in the GNPS datasets are not annotated, and they could correspond to either novel or known molecules. These spectra are searched against NIST17 by spectral library search and against AllDB by molDiscovery and Dereplicator+.

We compared the annotation rate (Supplementary Table 2), number of unique compounds (Fig. 5) and most annotated compounds (Supplementary Table 3) between spectral library search baseline and molDiscovery. Our results show that as expected, molDiscovery can annotate more tandem mass spectra than spectral library search. The top 10 compounds identified by molDiscovery at different mass ranges are shown in Supplementary Table 4.

At the strict 0% FDR level, molDiscovery annotated eight times more spectra (56,971 versus 7684, see Supplementary Fig. 17) and identified six times more unique compounds (3185 versus 542, see Fig. 5b) than Dereplicator+.

MolDiscovery search took 34 days on 10 threads, which is very close to the projected 329 days on a single thread. CSI:FingerID, CFM-ID, Metfrag and MAGMa+ are not benchmarked, as the searches would have taken years (see projections in Supplementary Fig. 18). It is worth noting that when searching such large-scale spectral datasets, methods that require preprocessing (molDiscovery and CFM-ID) are much more efficient than others, as they only need to preprocess the molecule database once, which can be used for searching any future spectra efficiently. The long running times of CSI:FingerID and MetFrag are probably due to the inefficient combinatorial algorithms.

**Scalability of molDiscovery to large molecular databases**. To demonstrate molDiscovery performance in high-throughput analysis of general metabolites, we benchmarked its ability to identify high-resolution MoNA spectra against a combination of 1.3 million bioactive-PubChem compounds and the ground truth compounds of MoNA spectra.

The search against a database with more than a million compounds is prohibitively time-consuming. The projected running times for all considered approaches on the MoNA spectra are

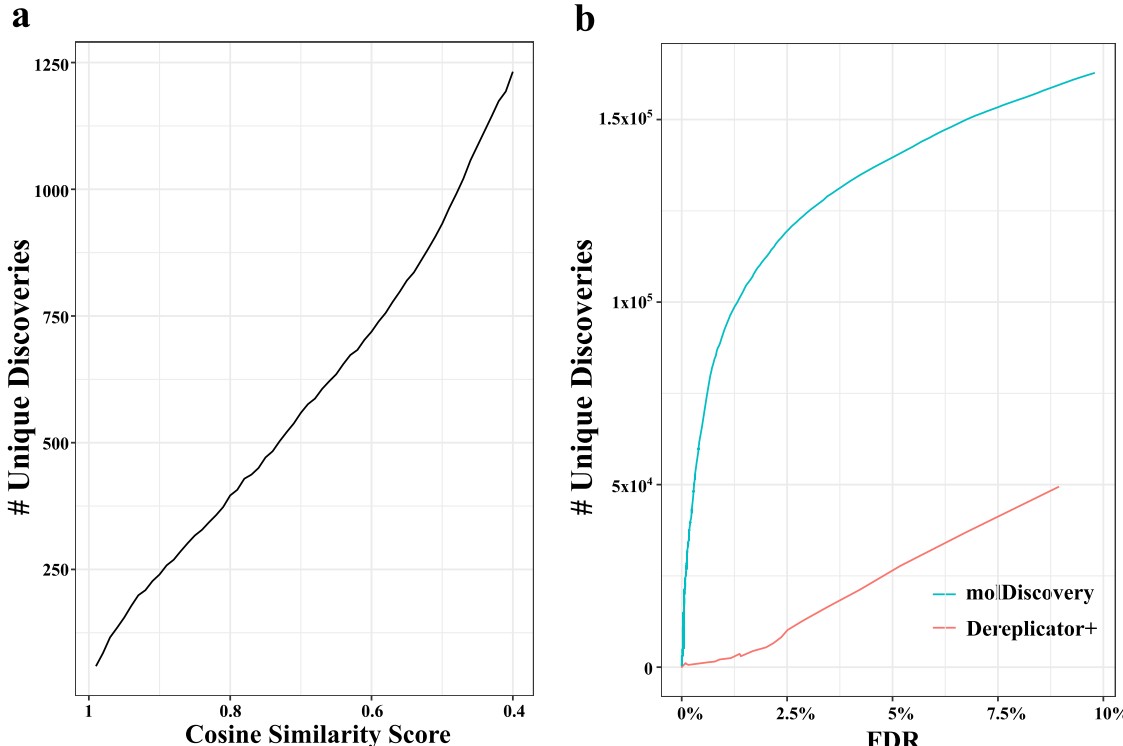

**Fig. 5 Number of unique discoveries by spectral library search and in silico database search. a** spectral library search against NIST17 and **b** in silico database search against AllDB.

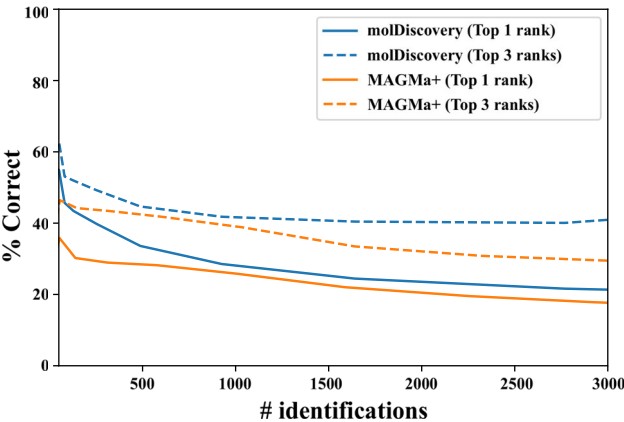

**Fig. 6 Accuracy of high-throughput identification of the MoNA mass spectra against bioactive-PubChem.** X-axis represents the number of identifications. Y-axis shows the percentage of correct identifications among the top identifications.

shown in Supplementary Table 5. Due to resource limitations, we analyzed this dataset only with molDiscovery and MAGMa+, the fastest and the most memory-efficient tools among the competitors. We ran both tools on a server with 20 CPUs Intel Xeon 2.50 GHz. The real execution times of molDiscovery and MAGMa+ were 38 min and 1988 min, respectively, which are in line with the single-thread projections in Supplementary Table 5. Fig. 6 shows results of molDiscovery and MAGMa+ for top 1 and top 3 ranked identifications. Supplementary Fig. 19 shows the detailed results for multiple ranks and various score thresholds. MolDiscovery demonstrates better or the same accuracy as MAGMa+ while achieving the results significantly faster.

**Validating molDiscovery identifications using a literature search.** We benchmarked molDiscovery against Dereplicator+ on top 100 identifications from GNPS datasets by performing literature search. For this task, we used the extensively studied GNPS dataset MSV000079450 (~400,000 spectra from *Pseudomonas* isolates)[41,42]. Out of the top 100 small molecule-spectra matches reported by molDiscovery, 78 correspond to compounds having *Pseudomonas* origin based on taxonomies reported for molecules in the AntiMarin database. The second largest genus among the identifications (20 out of 100) is *Bacillus*. The molecule–spectrum matches from *Bacillus* are likely to be true positives as the dataset is known to be contaminated with *Bacillus* species[42]. While the top 100 identifications from Dereplicator+ also contains 20 *Bacillus* matches, the number of hits related to *Pseudomonas* species is 62, 25% lower than molDiscovery. Eighteen identifications of Dereplicator+ are annotated as having fungi origin, which are likely false positives.

**MolDiscovery links secondary metabolites to their biosynthetic gene clusters (BGCs).** We cross-validated molDiscovery and genome mining by searching 46 microbial datasets (922 unique microbial strains, 8,013,433 spectra) that contained both tandem mass spectrometry data and genomic data. MolDiscovery successfully identified 19 molecules of various categories in microbial isolates that contained their known BGCs (Table 2), including 5 nonpeptidic molecules and 9 molecules <600 Da.

Moreover, molDiscovery successfully discovered novel BGCs for three small molecule families from *Streptomyces* dataset MSV000083738[43,44]. MolDiscovery search results for this dataset are available at https://github.com/mohimanilab/molDiscovery. MolDiscovery identified dinghupeptins A–D[45] in multiple *Streptomyces* strains, including *Streptomyces* sp. NRRL B-5680,

*S. varsoviensis* NRRL B-3589, *Ampullariella violaceochromogenes* NRRL B-16710 and *S. californicus* NRRL B-3320. AntiSMASH[46] revealed a non-ribosomal peptide (NRP) BGC in the genome of *Streptomyces* sp. NRRL B-5680, with seven adenylation domains that are highly specific to the amino acid residues of dinghupeptin molecules[47] (Fig. 7a). After a BLAST search of this BGC against the NCBI nucleotide collection database[48] and the MIBiG database, the maximum coverages are 37% and 20% respectively, indicating this NRP BGC is not similar to any known BGC.

MolDiscovery detected lipopeptin A–C in multiple *Streptomyces* strains, including *S. hygroscopicus* NRRL B-1477, *S. rimosus* NRRL WC3874, *S. rimosus* NRRL WC3904 and *S. rimosus* NRRL B-2659. Moreover, neopeptin A-C are identified in *S. hygroscopicus* NRRL B-1477, *S. rimosus* NRRL WC387, *Streptomyces* sp. NRRL S-1824 and *Streptomyces* sp. NRRL WC3549. Genome mining revealed an NRP BGC in *S. hygroscopicus* NRRL B-1477 with seven adenylation domains, which are highly specific to the residues of neopeptin and lipopeptin (Fig. 7b). It has been reported that these two families are structurally similar[49]. Thus, this BGC could be responsible for the production of both families. BLAST results show the maximum coverages of this BGC by known sequences are only 25% and 15% in the NCBI nucleotide collection database and the MIBiG database.

In addition, molDiscovery identified longicatenamycin family of cyclic hexapeptides in *S. rimosus* NRRL WC3558, *S. rimosus* NRRL WC3925, *S. rimosus* NRRL B-8076, *S. lavendulae* NRRL B-2775, *S. rimosus* NRRL B-2661, *S. griseus subsp. rhodochrous* NRRL B-2931 and *S. rimosus* NRRL WC-3930. Genome mining of the *S. rimosus* NRRL B-8076 strain revealed a BGC highly specific to longicatenamycin (Fig. 7c). Dereplicator+ and CSI: FingerID failed to discover longicatenamycin at 1% FDR threshold. BLAST results show neither the NCBI nucleotide collection database nor the MIBiG database have known genes homologous to this BGC.

## Discussion

With the advent of high-throughput mass spectrometry, large-scale tandem mass spectral datasets from various environmental/clinical sources have become available. One of the approaches for annotating these datasets is in silico database search. Currently the existing methods for in silico database search of tandem mass spectra can not scale to searching molecules heavier than 1000 Da and fail to explain many of the fragment peaks in tandem mass spectra.

MolDiscovery introduces an efficient algorithm to construct fragmentation graphs of small molecules, enabling processing molecules up to 2000 Da and molecular databases with large number of molecules. MolDiscovery is one of the most efficient methods in terms of both searching running time and memory consumption, which make it scalable to search of large-scale spectral datasets.

Furthermore, by training on the reference spectra from the GNPS spectral library, molDiscovery learned a probabilistic model that outperforms the other methods in overall accuracies, various mass ranges and compound classes. While molDiscovery is the best performing method in high mass range (>600 Da), similar to other database search methods, it has lower accuracy for molecules in low mass ranges (<600 Da). This is partially because fewer fragments are generated for molecules in this range. This makes it crucial to increase the accuracy of fragmentation models of small molecules.

Mass spectrometry fragmentation is influenced by experimental factors such as instrument type, dissociation technique, ionization mode, collision energy, etc. Currently, the default version of molDiscovery is trained on a combination of CID and

**Table 2 MolDisovery identified 19 molecules in bacterial isolates that contained their known BGCs.**

| molecule | Type | Mass (Da) | MiBiG ID | Organism | taxa similarity | GenBank ID | MSV ID | Blast Coverage | FDR (%) | Score |
|---|---|---|---|---|---|---|---|---|---|---|
| xantholysin A | NRP | 1775.08 | BGC0000463 | P. mosselii BW11M1 | species | GCA_001562525.1 | MSV000085018 | 1 | 0 | 171.16 |
| xantholysin B | NRP | 1761.07 | BGC0000463 | P. mosselii BW11M1 | species | GCA_001562525.1 | MSV000085018 | 1 | 0 | 141.45 |
| Orfamide C | NRP | 1266.81 | BGC0000399 | P. protegens Pf-5 | strain | GCA_000012265.1 | MSV000085018 | 1 | 0 | 130.09 |
| xantholysin C | NRP | 1801.1 | BGC0000463 | P. mosselii BW11M1 | species | GCA_001562525.1 | MSV000085018 | 1 | 0 | 121.87 |
| Orfamide A | NRP | 1294.84 | BGC0000399 | P. protegens Pf-5 | strain | GCA_000012265.1 | MSV000085018 | 1 | 0 | 116.64 |
| hygromycin A | Saccharide | 511.169 | BGC0000698 | S. hygroscopicus subsp. hygroscopicus NRRL B-1477 | species | GCA_000721535.1 | MSV000083738 | 0.95 | 0 | 76.3 |
| Orfamide B | NRP | 1280.82 | BGC0000399 | P. protegens Pf-5 | strain | GCA_000012265.1 | MSV000085018 | 1 | 0 | 75.54 |
| chlortetracycline | PK | 478.114 | BGC0000209 | Streptomyces sp. NRRL B-5680 | family | GCA_000719415.1 | MSV000083738 | 0.96 | 0 | 64.05 |
| oxytetracycline | PK | 460.148 | BGC0000254 | S. rimosus subspecies rimosus NRRL B-2626 | species | GCA_000721045.1 | MSV000083738 | 1 | 0 | 60.11 |
| PK-1 | PK/NRP | 342.252 | BGC0000972 | E. coli IHE3034 | species | GCA_000025745.1 | MSV000078995 | 1 | 0 | 51.13 |
| nostopeptolide A | PK/NRP | 1066.57 | BGC0001028 | N. punctiforme PCC 73102 | genus | GCA_000020025.1 | MSV000078891 | 0.98 | 0 | 50.72 |
| PK-2 | PK/NRP | 439.341 | BGC0000972 | E. coli IHE3034 | species | GCA_000025745.1 | MSV000078995 | 1 | 0 | 49.17 |
| PK-3 | PK/NRP | 340.236 | BGC0000972 | E. coli IHE3034 | species | GCA_000025745.1 | MSV000078995 | 1 | 0 | 48.03 |
| plipastatin | NRP | 1462.8 | BGC0000407 | B. subtilis subsp. subtilis 168 | species | GCA_000009045.1 | MSV000084117 | 1 | 0 | 43.44 |
| CDA4a | NRP | 1494.52 | BGC0000315 | S. coelicolor A3(2) | strain | GCA_000203835.1 | MSV000078839 | 1 | 0 | 43.24 |
| Xenocoumacin 1 | PK/NRP | 465.259 | BGC0001054 | X. miraniensis DSM 17902 | genus | GCA_002632615.1 | MSV000081063 | 0.9 | 0 | 41.47 |
| candicidin | PK | 1108.57 | BGC0000034 | Streptomyces sp. CNY228 | genus | GCA_000377545.1 | MSV000078839 | 1 | 0.01 | 32.43 |
| yersiniabactin | NRP | 481.116 | BGC0000467 | E. coli MS 21-1 | order | GCA_000164355.1 | MSV000082045 | 1 | 0.01 | 31.12 |
| sangivamycin | other | 309.107 | BGC0000879 | S. rimosus NRRL WC3904 | species | GCA_000720725.1 | MSV000083738 | 0.95 | 0.01 | 26.96 |

A genome is reported to contain a known BGC from the MIBiG database[35] if the genome has total BLAST hit[48] to the BGC with at least 90% coverage and 70% identity. Taxa similarity means similarity between the taxon of source organism and the organism reported in MIBiG. NRP stands for non-ribosomal peptides, PK stands for polyketides, PK-1 stands for N-myristoyl-D-asparagine, PK-2 stands for (R)-N1-((S)-5-oxohexan-2-yl)-2-tetradecanamidosuccinamide, and PK-3 stands for cis-7-tetradecenoyl-D-asparagine.

HCD spectra. While molDiscovery also includes pretrained models consisting of CID only and HCD only spectra, our results show that these models do not outperform the default model on CID or HCD datasets. Moreover, while the default parameters are trained on positive mode spectra, molDiscovery still performs well on negatively charged spectra. Since molDiscovery supports training by custom spectral datasets, users can adapt the default model to their experimental settings by training on in-house data.

Small molecule fragmentation is a complex process that depends not only on the type of fragmented bonds but also on local/global features of small molecules such as moiety. Currently, molDiscovery ignores these features and only covers a limited number of bond types and rearrangement rules, e.g., molDiscovery naively treats benzene ring or large resonant structures as alternate single bonds and double bonds. This could be improved by incorporating more complex fragmentation and rearrangement rules. Recent advances in graph-based machine learning have enabled representing complex small molecule structures with continuous vectors, making it feasible to incorporate local/global structural information into the prediction of fragmentation, potentially leading to more accurate fragmentation models. MolDiscovery paves the way for these more sophisticated approaches by collecting larger training data of small molecule-spectrum matches through the search of millions of spectra.

MolDiscovery computes the scores between molecules and mass spectra based on the log-likelihood ratio of a probabilistic model. The higher the score is, the more likely the spectrum is generated by the molecule than by random chance. In addition, molDiscovery estimates FDR using the target-decoy approach. While a Markov Chain Monte Carlo algorithm for computing p-value of molecule-spectrum matches in the case of peptidic molecules has been developed, extending this approach to general molecules is challenging as it involves generation of random molecular structures.

## Methods

**Constructing fragmentation graphs of small molecules.** MolDiscovery first constructs a metabolite graph for a small molecule structure and then generates a fragmentation graph from the metabolite graph (Fig. 8). To simplify the modeling of small molecule fragmentation, we assume that mass spectrometry can only break N-C, O-C, and C-C bonds. This is a reasonable assumption, as among top nine most frequent bonds in AntiMarin, these three bonds are the only single bonds that do not contain a hydrogen atom (Supplementary Table 9). Currently, we naively regard benzene ring and large resonant structures as alternate carbon-carbon single bonds (C-C) and double bonds (C=C). As molDiscovery can only cut at C-C, C-N, and C-O bonds, it will only cut single C-C bonds in benzene rings. The rearrangement rules associated with the bond fragmentation are listed in Supplementary Table 10.

To construct a metabolite graph, molDiscovery first disconnects N-C, O-C, and C-C bonds. The resulting connected components form the nodes of the metabolite graph. Edges in the metabolite graph correspond to bonds between the connected components (Fig. 8a, b).

The fragmentation graph of a molecule is a directed acyclic graph with a single source (the metabolite graph) where nodes are fragments of the molecule and directed edges between the nodes represent bridge or 2-cut fragmentations. To construct a fragmentation graph, molDiscovery first searches for all the bridges and 2-cuts of the metabolite graph to obtain depth one fragments of the molecule using Hopcroft and Tarjan's algorithm[50] for graph manipulation. Each fragment of the molecule can be represented by a fixed length binary vector that indicates the presence or absence of metabolite graph nodes in the fragment (Fig. 8d). We observe that depth two fragments can be formed by a bitwise *AND* operation between their parent fragments and depth one fragments (Fig. 8e). This generalizes to computing fragments at depth $n > 1$ by intersecting their parents (fragments at depth $n-1$) with fragments at depth one. The final fragmentation graph is constructed by connecting the source node to all depth one fragment nodes, and then iteratively connecting depth $n-1$ fragment nodes to the corresponding fragments at depth $n$ (Supplementary Note 2).

**Learning a probabilistic model to match mass spectra and small molecules.** Dereplicator+ uses a naïve scoring scheme which ignores that (i) fragmentation probability of different bonds are different, (ii) bridges have a higher chance of fragmentation than 2-cuts, (iii) peaks with higher intensity have higher chances of

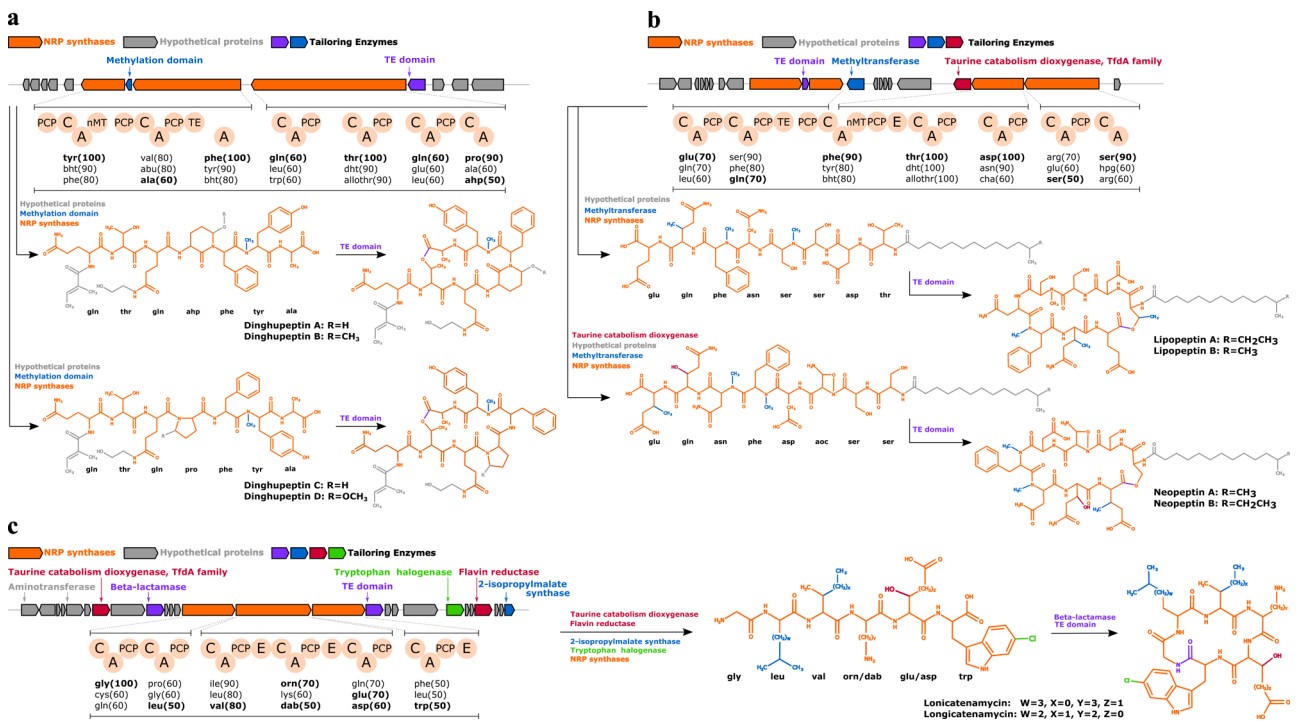

**Fig. 7 Putative BGCs of three secondary metabolite families identified by molDiscovery. a** dinghupeptin family in *Streptomyces* sp. NRRL B-5680, **b** lipopeptin and neopeptin family in *S. hygroscopicus* NRRL B-1477 and **c** longicatenamycin family in *S. rimosus* NRRL B-8076. MolDiscovery identified four dinghupeptin variants, two lipopeptin variants, neopeptin variants and two longicatenamycin variants at FDR 0%. After searching the corresponding genomes using antiSMASH, we detected NRP BGC with adenylation domains which show high specificity to the amino acids residues of these families. We also used DFAST (Supplementary Table 6–8) and HMMsearch to annotate the genes in the gene cluster. The post-modifications and the corresponding putative tailoring enzymes are color coded. In the putative BGC of longicatenamycin family, it is known that hydroxylation (red) can be induced by the Taurine catabolism dioxygenase[57], and flavin reductase can involve in FAD-dependent hydroxylation[58]. 2-isopropylmalate synthase (blue) can function on the isopropyl group and elongate the chain (EC 2.3.3.13).

corresponding to a fragmentation than lower intensity peaks, (iv) matches are biased towards molecules with large fragmentation graph size, and (v) matches are biased towards spectra with a large number of peaks.

In order to solve the above shortcomings we develop a probabilistic model for matching spectra with small molecules. Given a molecule-spectrum pair in the training data (Fig. 1a, b), we first construct the fragmentation graph of the molecule using our fragmentation graph construction algorithm (Fig. 1c). Each fragment in the fragmentation graph is assigned a (*bondType*, *logRank*) label. *bondType* represents the bond(s) disconnected in the parent fragment to produce the current fragment. It can either be one bond (bridge) or two bonds (2-cut). Bridges can be OC, NC, CC, while 2-cuts can be their pairwise combinations.

*logRank* represents the intensity of the mass peak corresponding to the fragment (Fig. 1c, d). The mass spectrometry community has used intensity rank as an abundance measure of mass peaks in a spectrum[51,52]. The higher the intensity rank is (closer to rank 1), the more abundant is the corresponding fragment. To reduce the number of parameters and avoid overfitting, we group peaks according to their *logRank* instead of directly using intensity rank (Supplementary Note 3). A fragment will be annotated with a *logRank* between 1 and 6 if there is a peak with rank between 1 and 64 in the spectrum within 0.01 Da of the mass of the fragment (Fig. 1cd). If there is no such mass peak, the fragment will be annotated with *logRank* = 7.

In the annotated fragmentation graph (Fig. 1c) we assume that (i) the *logRank* of each fragment depends only on its *bondType* and the *logRank* of its parent. Here, we only consider direct parent, as considering grandparents of the fragments increases the number of parameters by an order of magnitude, resulting in overfitting. We further assume (ii) *logRank* of each mass peak is independent from the *logRank* of other peaks. While this assumption is sometimes wrong, e.g., only one peak can have *logRank* 1, we use this assumption to simplify our probabilistic model. Finally, we assume (iii) the root node has *logRank* 0. Given a small molecule structure *Molecule* and its fragmentation graph *FG*, the probability of generating a spectrum *Spectrum* is as follows:

$$P(Spectrum|Molecule) = \prod_{peak \in Spectrum} P(peak|FG) \tag{1}$$

$$= \prod_{peak \in Spectrum_{FG}} P(peak|FG) \prod_{peak \in Spectrum_{null}} P(peak|FG) \tag{2}$$

where *Spectrum_FG* represents all the peaks within 0.01 Da of some fragment in *FG*, and *Spectrum_null* represents the rest of the peaks. If multiple fragments match the same peak, we randomly pick one with lowest depth. Since we use *logRank* as a measure of abundance and by definition all peaks in *Spectrum_FG* correspond to a fragment in *FG*, we can rewrite the equation as follows:

$$P(Spectrum|Molecule) = \prod_{frag \in FG} P(logRank_{frag}|FG) \prod_{peak \in Spectrum_{null}} P(peak|FG) \tag{3}$$

where *frag* is a fragment in the fragmentation graph. Then, by assumption (i):

$$P(Spectrum|Molecule)$$
$$= \prod_{frag \in FG} P(logRank_{frag}|bondType_{frag}, logRank_{pa(frag)}) \prod_{peak \in Spectrum_{null}} P(peak|FG) \tag{4}$$

where *pa(frag)* is the parent of *frag*. Similarly, we can obtain the probability of generating a random spectrum (*null* model):

$$P(Spectrum|null) = \prod_{peak \in Spectrum} P(peak|null) \tag{5}$$

In order to learn $P(logRank_{frag}|bondType, logRank_{pa(frag)})$, we can directly count the number of the fragments with a particular *logRank* for each (*bondType = b*, *logRank_{pa} = p*) combination in the training data as follows:

$$P(logRank = l|bondType = b, logRank_{pa} = p)$$
$$= \frac{\sum_i |\{frag \in FG_i | logRank_{frag} = l, bondType_{frag} = b, logRank_{pa(frag)} = p\}|}{\sum_i |\{frag \in FG_i | bondType_{frag} = b, logRank_{pa(frag)} = p\}|} \tag{6}$$

where $FG_i$ is the fragmentation graph of the *i*th molecule-spectrum pair in the training data. Similarly, for the null model, we can compute the *logRank* distribution of all the mass peaks as follows:

$$P(logRank = l|null) = \frac{\sum_i |\{peak \in Spectrum_i | logRank_{peak} = l\}|}{\sum_i |\{peak \in Spectrum_i\}|} \tag{7}$$

where $Spectrum_i$ is the spectrum of the *i*th molecule-spectrum pair.

**Scoring a spectrum against a small molecule.** Given a query tandem mass spectrum (Fig. 1h) and a small molecule in a chemical structure database (Fig. 1g),

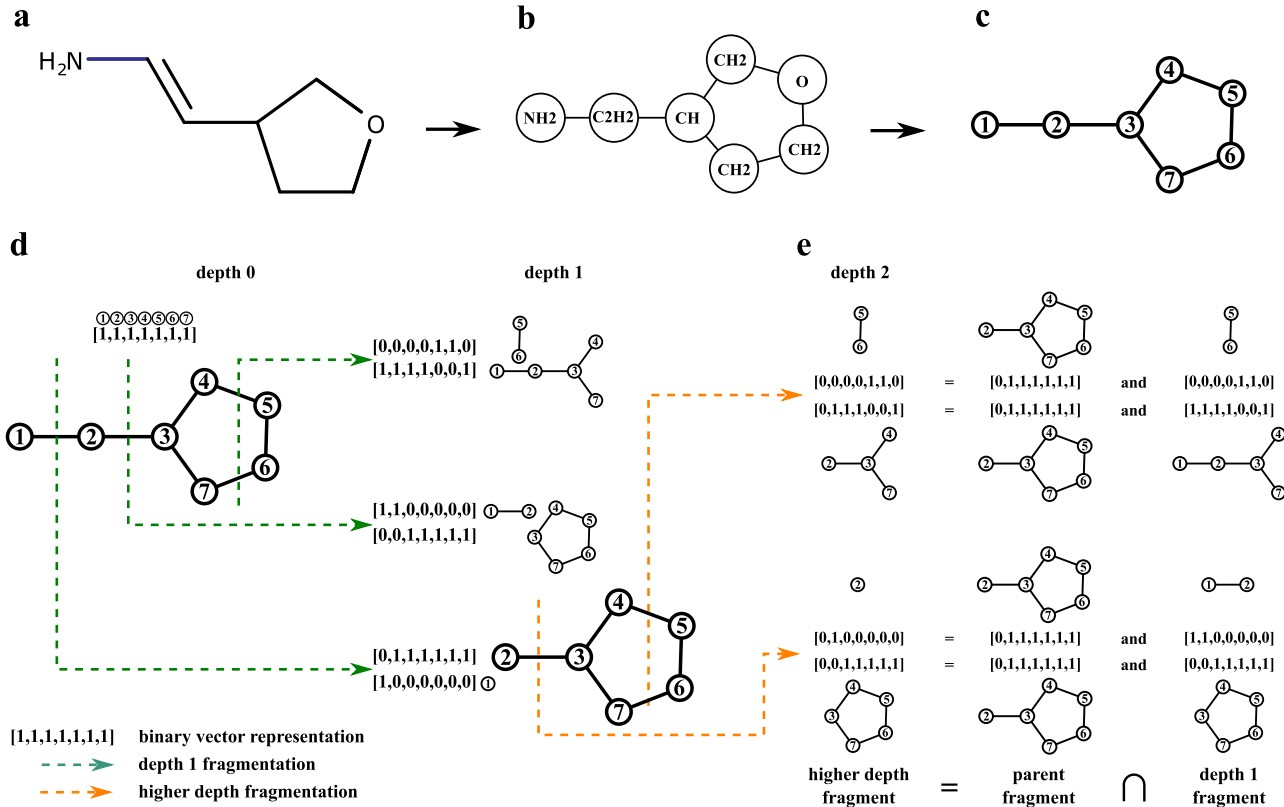

**Fig. 8 Overview of fragmentation graph construction algorithm. a** A small molecule. **b** Metabolite graph of the molecule. **c** Indexed metabolite graph. **d** Construction of the fragmentation graph of the small molecule. We assign an index to each component of a molecule and represent each of its fragments as a binary vector, where 1 and 0 indicate presence and absence of the components. Each bridge/2-cut of a parent fragment gives two child fragments. **e** Higher depth fragments can be computed as overlaps of their parents with the depth one fragments.

if the spectrum precursor mass is within 0.02 Da of the mass of the small molecule and the user-specified adduct, they will become a candidate pair for scoring. We then construct the fragmentation graph of the molecule and annotate the query spectrum against the fragmentation graph (Fig. 1i, j). Based on the described probabilistic model, we use a log-likelihood ratio model (Fig. 1k) to score the spectrum against the small molecule:

$$\log \frac{P(Spectrum|Molecule)}{P(Spectrum|null)}$$

$$= \log \prod_{peak \in Spectrum_{FG}} \frac{P(peak|FG)}{P(peak|null)} \prod_{peak \in Spectrum_{null}} \frac{P(peak|FG)}{P(peak|null)} \quad (8)$$

$$= \log \prod_{frag \in FG} \frac{P(logRank_{frag}|FG)}{P(logRank_{frag}|null)} \quad (9)$$

$$= \log \prod_{frag \in FG} \frac{P(logRank_{frag}|bondType_{frag}, logRank_{pa(frag)})}{P(logRank_{frag}|null)} \quad (10)$$

$$= \sum_{frag \in FG} \log \frac{P(logRank_{frag}|bondType_{frag}, logRank_{pa(frag)})}{P(logRank_{frag}|null)} \quad (11)$$

Note that in (9), we only need to consider the mass peaks corresponding to fragments in the fragmentation graph since we assume all the other peaks are generated by random chance, hence $\frac{P(peak|FG)}{P(peak|null)} = 1$.

**Computing FDR**. We use target-decoy analysis to estimate FDR[53]. First, we randomly shuffle the fragmentation graph of target molecules to create decoy fragmentation graphs. Then, tandem mass spectra are searched against the target and the decoy fragmentation graph databases respectively. At each score cutoff, there are $N_{target}$ matches to the target database and $N_{decoy}$ matches to the decoy database, and the FDR is estimated as

$$FDR = \frac{N_{decoy}}{N_{target}}$$

Supplementary Fig. 20 shows an example of $N_{target}$ and $N_{decoy}$ calculation at different score thresholds for a search of a *Streptomyces* dataset.

## Data availability
All the data supporting the findings of this study are available within the Article and Supplementary Information, or are deposited in Zenodo[54].

## Code availability
The command-line version of molDiscovery is available at https://github.com/mohimanilab/molDiscovery. The online web service through the GNPS infrastructure are available at https://gnps.ucsd.edu/ProteoSAFe/index.jsp?params=%7B%22workflow%22:%22MOLDISCOVERY%22%7D. Please also see Supplementary Note 4 for instructions of running molDiscovery on GNPS.

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

## Acknowledgements

The work of L.C., M.G., Y.L., and H.M. was supported by a research fellowship from the Alfred P. Sloan Foundation, a National Institutes of Health New Innovator Award DP2GM137413, and a U.S. Department of Energy award DE-SC0021340. The work of A.T. and A.G. was supported by RFBR (project number 20-04-01096). Research was carried out in part by computational resources provided by the Computer Center of Research park of St. Petersburg State University. Also, this work used the Extreme Science and Engineering Discovery Environment (XSEDE), which is supported by National Science Foundation grant number ACI-1548562. Specifically, it used the Bridges system, which is supported by NSF award number ACI-1445606, at the Pittsburgh Supercomputing Center (PSC)[55,56].

## Author contributions

L.C. took the lead in implementing model and writing manuscript. M.G., A.T., and A.G. improved the software and performed benchmarking experiments. Y.L. analyzed secondary metabolites data. H.M. supervised the project. All authors helped shape the research, analysis and manuscript.

## Competing interests

H.M. is a co-founder and have equity interest from Chemia.ai, LLC. The other authors declare no competing interests.
