## [Peer Review File · Nature Communications]

Reviewers' Comments:

Reviewer #1:

Remarks to the Author:

The authors tackle a timely subject in the field of metabolomics, being the identification of small molecules via predicted fragmentation patterns. In their manuscript they describe the MolDiscovery tool to predict fragments from a wide variety of small molecules. Although I believe that the tool is of interest, major revisions would be needed to reach impact at the broad community of metabolomics researchers.

General/Major Comments:

- 1) The current focus of the manuscript lies in the prediction of fragmentation patterns of metabolites which can be used for the improved identification of metabolites. I believe that focus should be on the latter, with representative databases (MSMS spectra) that cover the majority of metabolomics research. For this, datasets covering polar metabolites (range 50-750 Da) and larger apolar species (lipids) will trigger the interest of many groups in the field. Thus, I believe that the authors should validate the MolDiscovery tool on datasets from human serum (metabolites/lipids).
- 2) page 5: many of the polar compounds are also phosphorylated, why are these bondTypes not covered in the predicted fragments?
- 3) What input files does the online tool require? It is not stated and in addition, has the compatibility been checked with the major MS providers? (being Waters, Agilent, Thermo, Bruker, ABSciex) It is also not stated for which MS instruments this tool would be useful or could be used. Ideally, the input file is of the type mzML which would cover the majority of MS providers.
- 4) the online tool itself does not provide the option to include a relative error on the parent mass (it is only in absolute values for now). Since the majority of MS instruments have gone beyond this, capable of measuring the parent mass with an accuracy <10 ppm, it would be good to have this included. Also, how does the exact parent mass then influence the scoring of metabolites?
- 5) it would be convenient for researchers to have the ability to indicate the level of confidence regarding the identifications (95% confidence, 99%confidence etc)
- 6) for the output, in metabolism the software cannot discriminate between isobaric species (think of leucine/isoleucine) which often co-elute. For this, having a list of potential 'overlapping' species would be needed.
- 7) figure 7: the third group from the left states "...non-mental compounds" I believe this is a typo?
- 8) can the tool be used to predict fragmentation of small molecules by diverse fragmentation modes? For instance ETD, CID, HCD,
- 8) can the tool deal with batch searches from multiple MSMS files?
- 9) similar to other prediction tools, the performance is affected by the molecular size of the target. Especially in the low m/z range, all tools seem to struggle (probably because less fragments are generated), unfortunately this is the among the most interesting range for metabolomics research. Can the authors comment on this?

Reviewer #2:

Remarks to the Author:

Review of Ms. No.: NCOMMS-20-35721

The current article focuses on a major challenge in molecular mass spectrometry – and moves the needle in this populated and venerable area. The work advances the implementation and scoring (including control of FDR for large datasets) of taking moderate and high resolution tandem mass spectrometric datasets and converting those into compound assignments from a database of tens of thousands of knowns. Further, the ability to assert detection of analogs of known compounds is also shown; this is akin to error-tolerant database searching in mass spec-based proteomics. Another application demonstrated is to detect natural products, with some examples of assignment to their BGCs. This is sophisticated algorithmic performance, and therefore, publication is recommended, with a few provisos provided in the following Major and Minor comments.

Major:

Does the article support the central claims? Yes, mostly.

Claim 1: "Our results show that molDiscovery greatly increases the accuracy of small molecule identification, while making the search an order of magnitude more efficient."

- Supporting evidence is shown in the manuscript; however this reviewer did not validate that the code base downloads, compiles and functions as advertised (e.g., generates the FDR estimation that shows better performance as shown in the manuscript).

Claim 2: "Moreover, by applying molDiscovery on microbial datasets with both mass spectral and genomics data we successfully discovered the novel biosynthetic gene clusters of three families of small molecules."

- The Abstract-level claim needs to be tempered down. The 3 "novel" BGCs assigned to the compounds need to be carefully evaluated. These 3 examples are from known compound families and each one should be the result of a careful literature search to see if these assignments are indeed novel. In any case, they are "putative" assignments, and the results should be reported more carefully. In how many strains are these compounds observed? How are the masses related to those for known compounds in these molecular families? Supplemental Figures S7 and S8 can be augmented to capture this information, and some if it should be reported in textually the Main Text. If indeed these assignments of known compounds (and their analogs) to their BGCs are all truly novel – that is a major accomplishment for the work.

Minor:

- Suggest authors consider putting ~1-3 figures into the Supplemental Information, e.g., for algorithm validation.

- Discussion: "MolDiscovery is shown to be robust against dissociation technique" . Suggest rewording to be clearer to more general readers. Also, there are many dissociation techniques in mass spec., so this statement should be better justified/curtailed.

Reviewer #3:

Remarks to the Author:

The current paper discusses a software that can help in compound identification. The annotation of small molecules in complex matrices is an ongoing challenge, better algorithms are urgently needed. The MolDiscovery software described here is available as source code and also online via the GNPS database in a workflow format. The algorithms is very fast and has a small memory footprint, which is exciting.

The tool does not consider negative ionization mode spectra, which are important in metabolomics. This could be a future improvement. Many of the showcases are peptides or peptide-analogs and are very important but do not represent the full natural product space. Based on the limited and small benchmark set the claim that the tool works well for "small molecule fragmentations" has to be debated unless the authors show further proof. This proof could be easily done by including benchmarks from larger spectral sets including NIST or MassBank. The pseudomonas application is interesting, but not really convincing, here wider coverage should be applied including several mammalian, plant or even marine algal samples.

Major issues:

1) Abstract: "The existing approaches use the domain knowledge from chemistry to predict fragmentation of molecules." That is actually not true. See QCEIMS by Grimme 2013 which is a quantum chemical process to create fragmentations of small molecules (70 eV).

2) Section: "The majority of in silico database search methods are rule-based models that incorporate domain knowledge such as bond types, hydrogen rearrangement, and dissociation energy [16, 17, 18,19, 20, 21] to predict fragmentations of small molecules and score small molecule-spectrum pairs. However, due to the complex rules involved in small molecule fragmentation, these methods are computationally insufficient for heavy small molecules. Moreover, these predictions, which are based on expert knowledge, fail to explain many of the

peaks in mass spectra.”

So here we have to deal with three problems. Problem-1: it would be worthwhile to name the most successful approaches (MetFrag, CFM-ID, CSI-FingerID, MS-Finder, ChemDistiller). Many of these tools have been successfully used in CASMI (small molecule) contests or had very elaborate validations on thousands of small molecules (ChemDistiller, CFM-ID, CSI-FingerID). So simple citations are not sufficient to explain why they are discarded.

Problem-2: It would be useful to define “heavy” with a specific molecular weight and then also proof this assertion in the manuscript. Many of the tools can handle molecules up to 1500 Dalton.

Problem-3: In many cases the assertion that they rely on “expert knowledge” or use only rule-based approaches is not true, CFM-ID, SF-Matching and CSI-FingerID and others are machine learning based, no expert knowledge required.

3) The authors claim the algorithm can be used for identifying metabolite / small molecules.

However, most figures shown in the paper are about large molecule (>800-1000 Da) and mostly peptides. Most metabolites / small molecules have very low molecular weight. For example, 96.5% of molecules in the KEGG database have mass <1000 Da

(<https://doi.org/10.1093/bioinformatics/btn603>). In the NIST Tandem Mass Spectral Library 2020 release, more than 98% of spectra have a molecular weight less than 1,000 Da, more than 93% spectra have molecular weight less than 600 Da. To have a general interest in metabolite / small molecules identification, more data/examples about analyzing small molecules (<600 Da) should be presented.

4) “MolDiscovery identified 56,971 small molecules at 0% false discovery rate (FDR),” Please add to supplement with reference, links and IDs. Make sure to create histograms with the most annotated compounds. It could be 10,000-times alanine, 10,000-times glucose etc. So the number 56,971 does not say anything about the performance. Also in this case please match against the latest small molecule databases such as NIST20 or Massbank. Compare the annotation rates for spectral analysis.

5) Dataset section: “The GNPS spectral library contains 8,964 small molecule-spectrum matches” Looking at the GNPS download link (<https://gnps-external.ucsd.edu/gnpslibrary>) That is actually not true. There are 16,610 records with unique SMILES and 8,374 records with InchiKeys. The intrinsic problem here is that out of 156,355 GNPS records (ALL_GNPS.msp) very few actually have annotated structures.

The MONA positive mode LC-MS/MS libraries has 91,774 unique records based on around unique 13,600 structures, but for a single given molecule, multiple MS/MS spectra with different collision energies or even based on different instruments or in-source voltages could be derived.

So simply filtering by unique structure is not sufficient. This ignores the multiple CID or HCD voltages which can lead to very different spectra. CFM-ID for example was trained on different CID voltages and correctly takes this into account. So here I recommend to retrain based on all MassBank, NIST or MONA spectra. If retraining is not required include extended validation.

6) The paper utilizes Dereplicator or Dereplicator+ as baseline, however some of the best algorithms for small molecule annotations, including MetFrag or ChemDistiller are not even included or benchmarked here, (only CSI-FingerID and CFM-ID). These tools were comprehensively validated in their publications and also participated in the CASMI contests. Dereplicator+ has not been evaluated in such a way or even benchmarked against the best tools available, except in this paper (or the dereplicator publication). I recommend to utilize at least one more (ChemDistiller or Metfrag) and perform a benchmark test on 10,000 NIST or MONA database structures. Filtering of compounds and spectra that were used during training will be difficult, but this benchmark set should give a good baseline.

7) The software does not discuss rearrangement reactions. However they are important in mass spectrometry and several tools including CFM-ID or MS-Finder integrate such rules.

Rearrangements are very common in molecules with small molecular weight. How does the algorithm deal with this? This goes back to my prior point that the MolDiscovery software needs to be properly benchmarked against long-standing and existing tools.

8) Figure 4: The CFM-ID 2.0 results are quite low which is inconsistent with the latest CASMI

contest data (also blinded molecules). Also MAGMA results are surprisingly high, which is inconsistent with the latest CASMI contests. Here maybe false parameter settings are responsible, all the input and output parameters for each of the benchmark sets need to be shared. That also includes input/output results and all raw tables. The ChemDistiller paper (ChemDistiller: an engine for metabolite annotation in mass spectrometry; *Bioinformatics*, 34(12), 2018, 2096–2102) which is one of the later and accurately performing algorithms also comes to the conclusion that MetFrag, CSI-FingerID and CFM-ID accuracy ranks between (76-88% in the TOP5). So I believe something is wrong with the reported data here in this report, at least for CFM-ID.

9) Dataset section: Filtering via Splash (spectral bases) should not be done, but rather filtering based on structures. Using SMILES for filtering could be error-prone, Inchikeys should be used for that case.

10) The used application dataset is MSV000079450 (pseudomonas, a gram-negative bacterium) is a rather minimal dataset if we consider hundreds and thousands of other organisms including those from mammalian or plant species. So here a few additional datasets such as human blood plasma or common plants (rice or Arabidopsis) would be useful to evaluate the tool. Also many of the annotated compounds are peptides, so I wonder what happened to the 10,000 small core metabolites from core pathways that are commonly annotated in bacteria. This is a major weakness and could be related to the algorithm or the dataset.

11) Lipids are important natural products but they are completely ignored, here a lipid dataset could show the true performance of the tools

12) Most of the compounds annotated in the supplement are peptides. That is great, but then it would be important to say very early in the article abstract, results and discussions, that the algorithm mainly works for peptide annotations. That is fine, but that is exactly what the paper and supplement data show.

13) The paper focused on limited (peptidic) bond types. I want to know the expansibility of the method to some other bond types. Will this decrease the accuracy significantly? It's interesting to see the trends.

14) The existing methods "cuts" the molecules arbitrarily, what about the aromatic system? For example, how to divide the benzene ring or large resonant structures?

15) The authors should benchmark MolDiscovery in more scenarios, for example, searching spectra from CASMI / MoNA / Metlin / NIST against PubChem / ChEMBL / other big molecular databases. This benchmark is very important to convince people that molDiscovery can be used for searching general metabolites / small molecules.

Minor issues:

1) Abstract: "This is a challenging task as currently it is not clear how small molecules are fragmented in mass spectrometry." I think rephrasing that to "complicated" or difficult would be better fitted.

2) Abstract: "Recently, spectral libraries with tens of thousands of labelled mass spectra of small molecules have emerged" Please rephrase. "labelled" in context of analytical chemistry usually means isotopically labelled. If you refer to metadata-enriched or annotated spectra, please say so.

3) Discuss negative ionization mode mass spectra (negative mode)

4) Figure 6, doubly charged species are important for peptides, but are relatively unimportant for small molecule approaches. The low 10-15% accuracy shows that. I would not include that figure.

5) Figure 7 is a very interesting figure, unfortunately very hard to read due to the small bars, here horizontal bars would be better. The two outliers for "non-mental" (please correct) and

hydrocarbons are probably due to the training set used. Also CFM-ID uses a different library for peptides (not shown in the graph).

6) Figure 8 please spell out MA, OT, TA – no idea what that means. Also consistent large caps/small caps would be nice. Also most of the compounds are peptides or analogs and are important but do not represent the full natural product space.

In the supplement the LogRank is defined as “ 2^{i-1} most intense peaks”, what does this mean? Why are the peaks are exponential? It is also better to give a clear definition for “null spectrum”.

Response to Reviewers

Reviewer #1

C1.1 The authors tackle a timely subject in the field of metabolomics, being the identification of small molecules via predicted fragmentation patterns. In their manuscript they describe the MolDiscovery tool to predict fragments from a wide variety of small molecules. Although I believe that the tool is of interest, major revisions would be needed to reach impact at the broad community of metabolomics researchers.

R1.1 Thank you for the suggestions on our paper – we have addressed your comments below. The changes are indicated in red in the revised manuscript and the Supplement.

Major comments

C1.2 The current focus of the manuscript lies in the prediction of fragmentation patterns of metabolites which can be used for the improved identification of metabolites. I believe that focus should be on the latter, with representative databases (MSMS spectra) that cover the majority of metabolomics research. For this, datasets covering polar metabolites (range 50-750 Da) and larger apolar species (lipids) will trigger the interest of many groups in the field. Thus, I believe that the authors should validate the MolDiscovery tool on datasets from human serum (metabolites/lipids).

R1.2 The focus of the manuscript is the identification of metabolites by *in silico* database search, which relies on the prediction of fragmentation patterns of metabolites. One advantage of the identification through fragmentation prediction is that mass spectral peaks become interpretable by the annotation of the corresponding fragments. Another advantage is that the model will not overfit the training data (See R3.10, Supplementary Fig. 6). Regarding the performance of molDiscovery on polar or non-polar molecules, we added Supplementary Fig. 11 as follows. MolDiscovery is the best tool for polar metabolites based on all polarity measures, while for non-polar metabolites, there is no tool that is consistently better than others.

Supplementary Figure 11: Database search accuracy of **a** polar and **b** non-polar compounds. Three polarity measures predicted by the RDKit package, including topological polar surface area (TPSA), molecular refractivity (MR), and partition coefficient (logP), are used to classify a compound as polar and non-polar. Here, if MR or TPSA is greater than median value of all the testing compounds, or logP is less than 1, then the compound is called polar.

In the main text, we added the following sentence accordingly to the subsection “MolDiscovery performance on different compound classes and mass ranges” of the Results section:

Moreover, we compared the database search accuracies of these tools on polar and non-polar metabolites. MolDiscovery is the best performing tool for searching polar metabolites across all the polarity measures, while for non-polar metabolites there is no tool that is consistently better than others (Supplementary Fig. 11).

We also added molDiscovery experiment on a human serum dataset (See R3.14) and a benchmarking on lipid identification (See R3.15). These experiments include the analysis of molDiscovery identifications in the HMDB (metabolites) and LMSD (lipids) chemical structure databases.

C1.3 page 5: many of the polar compounds are also phosphorylated, why are these bondTypes not covered in the predicted fragments?

R1.3 We did not cover such bondTypes mainly for three reasons. First, in AntiMarin and other natural product databases, the frequencies of phosphorus-related bond types such as P-C and O-P are less than 0.1% (Supplementary Table 8). Second, there is few training data in the spectral libraries for these bond types. Third, the hydrogen rearrangement for these bond types is currently not well understood.

Now, we have added a new subsection “**The effect of various bond types**” and Fig. 5 to the Results section to explore the influence of different bondTypes on the performance of molDiscovery. Our results show that the overall improvement by adding bond types such as O-P, P-C and S-C is negligible.

The effect of various bond types. We examined a subset of NIST20 that contains S-C, O-P, and P-C bonds, and tested how accuracy is affected by addition of all the combinations of these bond types on top of our default C-C, N-C, and O-C bonds. Although adding any of these bond types increases accuracy, the largest top 1 accuracy difference is less than 1% compared to the default. The impact of removing C-C, N-C, and O-C bonds on the accuracy is also examined, and the results show that molDiscovery accuracy significantly drops if any of these default bond types are removed (Fig. 5).

Figure 5: Top K ($K = 1, 3, 5, 10$) accuracy of molDiscovery on a subset of high-resolution NIST20 spectra from QTOF instrument types. The default setting denotes the C-C, N-C, and O-C bond types. The effect of removing bond types (no O-C, no N-C, no C-C) and adding new bond types (S-C & P-C, S-C & O-P & P-C, S-C & O-P, S-C, P-C, O-P & P-C, O-P) are explored. Removing any of our default bond types results in a severe drop in top 1 accuracy (8.3% for C-C, 29.2% for N-C). Addition of P-C, S-C and O-P bonds does not improve the overall accuracy.

We have also added the following sentences to the “Datasets” subsection.

We also selected 2,382 singly-charged NIST20 spectra from Q-TOF instruments using the same filtering steps.

In the future, we will explore more bond types and the corresponding hydrogen rearrangement. We have updated the following sentence in the Discussion section.

Currently, molDiscovery ignores these features and only covers a limited number of bond types and rearrangement rules, e.g. molDiscovery naively treats benzene ring or large resonant structures as alternate single bonds and double bonds. This could be improved by incorporating more complex fragmentation and rearrangement rules.

C1.4 What input files does the online tool require? It is not stated and in addition, has the compatibility been checked with the major MS providers ? (being Waters, Agilent, Thermo, Bruker, ABSciex) It is also not stated for which MS instruments this tool would be useful or could be used. Ideally, the input file is of the type mzML which would cover the majority of MS providers.

R1.4 MolDiscovery is compatible with .mzML, .mzXML and .mgf files. We have updated the manual of molDiscovery at <https://github.com/mohimanilab/molDiscovery> to clarify it. We also add guideline about how to use the online tool as Supplementary Note 2 (See Step 3, 3.1 and 3.2):

Supplementary Note 2. Run molDiscovery on GNPS

To run molDiscovery on GNPS, please visit <https://gnps.ucsd.edu/ProteoSAFe/index.jsp?params=%7B%22workflow%22:%22MOLDISCOVERY%22%7D> (see below). To run molDiscovery in the command line, please visit <https://github.com/mohimanilab/molDiscovery> for details.

Run molDiscovery on GNPS

- Step 1. Open a browser, go to GNPS and login
- Step 2. Go to molDiscovery workflow page
- Step 3. Select input spectra
- Step 4. Set up parameters in molDiscovery
- Step 5. Submit job and check email for notification

Step 1. Open a browser, go to GNPS and login

<https://gnps.ucsd.edu/>

GNPS: Global Natural Products Social Molecular Networking

MassIVE Datasets | Documentation | Forum | Contact

User: Pass: Sign in

Don't have an account? Register

Please Login to Analyze Data at GNPS

Login to Existing Account Register New Account

 GNPS

GNPS is a web-based mass spectrometry ecosystem that aims to be an open-access knowledge base for community-wide organization and sharing of raw, processed, or annotated fragmentation mass spectrometry data (MS/MS). GNPS aids in identification and discovery throughout the entire life cycle of data; from initial data acquisition/analysis to post publication.

Step 2. Go to molDiscovery workflow page

<https://gnps.ucsd.edu/ProteoSAFe/index.jsp?params=%7B%22workflow%22:%22MOLDISCOVERY%22%7D>

Workflow Selection

Search Protocol:

Title:

Workflow Description

MOLDISCOVERY

MolDiscovery is a mass spectral database search method that improves both efficiency and accuracy of small molecule identification by (i) utilizing an efficient algorithm to generate mass spectrometry fragmentations, and (ii) learning a probabilistic model to match small molecules with their mass spectra.

Find out more details on molDiscovery project page.

Basic Options View documentation

Input File: Precursor Ion Mass Tolerance:
Fragment Ion Mass Tolerance:

Advanced Options

Workflow Submission

Email me at

Step 3. Select input spectra

Workflow Selection

Search Protocol:

Title:

Workflow Description

MOLDISCOVERY

MolDiscovery is a mass spectral database search method that improves both efficiency and accuracy of small molecule identification by (i) utilizing an efficient algorithm to generate mass spectrometry fragmentations, and (ii) learning a probabilistic model to match small molecules with their mass spectra.

Find out more details on molDiscovery project page.

Basic Options View documentation

Input File: Precursor Ion Mass Tolerance:
Fragment Ion Mass Tolerance:

Advanced Options

Metabolite database (DB) selection

NB! If a Custom DB is provided (via File Selection or URL), the Predefined DB choice is ignored

Predefined DB:
Custom DB File: Custom DB URL:

Metabolite-Spectrum Scoring parameters

Max Charge:
Min Significant Score:

Workflow Submission

Email me at

Select input spectra

Step 3.1 Add input spectra to user directory

3.1.1 Add spectra by uploading

3.1.2 Add spectra from public MassIVE Datasets

Select Input Files

Upload Files

Share Files

Select Input Files

Selected Files

Selected Input File

Clear Selection

Finish Selection

CCMS_ProteomeDatabases

CCMS_School_2019

CCMS_SpectralLibraries

Guest

RMSV000000248

speclibs

Step 3.2 Select input spectra from user directory

3.2.1 Choose input spectra (all the mzML, mzXML and mgf files under the chosen directory will be searched)

3.2.2 Click to add

3.2.2 Click to finish selection

Select Input Files

Upload Files

Share Files

Select Input Files

Selected Files

Selected Input File

MSV000083738

Clear Selection

Finish Selection

CCMS_ProteomeDatabases

CCMS_School_2019

CCMS_SpectralLibraries

Guest

[Dataset MSV000083738] - "Full set golden dataset producing strains"

RMSV000000248

speclibs

Step 4. Set up parameters in molDiscovery

- Workflow Description:** MOLDISCOVERY description and project page link.
- Basic Options:** Input File (Select Input Files), View documentation, Precursor Ion Mass Tolerance (0.01 Da), Fragment Ion Mass Tolerance (0.01 Da).
- Advanced Options:** Metabolite database (DB) selection, Hide Fields.
 - Annotation: "Score threshold" points to the Min Significant Score field (10.0).
- Workflow Submission:** Email me at field, Submit button.

Step 5. Submit job and check email for notification

- Workflow Description:** MOLDISCOVERY description and project page link.
- Basic Options:** Input File (Select Input Files), View documentation, Precursor Ion Mass Tolerance (0.01 Da), Fragment Ion Mass Tolerance (0.01 Da).
- Advanced Options:** Metabolite database (DB) selection, Hide Fields.
 - Annotation: "Submit job" points to the Submit button.
- Workflow Submission:** Email me at field, Submit button.

C1.5 the online tool itself does not provide the option to include a relative error on the parent mass (it is only in absolute values for now). Since the majority of MS instruments have gone beyond this, capable of measuring the parent mass with an accuracy <10 ppm, it would be good to have this included. Also, how does the exact parent mass then influence the scoring of metabolites?

R1.5 If the parent mass of an experimental spectrum lies within the error range of the mass of a small molecule in the database (after taking into account the possible isotopic shifts and adducts), it will be scored against the small molecule. Otherwise, no score will be computed.

We have added the option of relative error of the parent mass to the online tool. Now both the command-line version and the online version have the relative error option. We also added guidelines about how to use the online tool to Supplementary Note 2 (See Step 2). See R1.4.

C1.6 it would be convenient for researchers to have the ability to indicate the level of confidence regarding the identifications (95% confidence, 99%confidence etc)

R1.6 Currently, we do not have a method for p-value or confidence interval computation in molDiscovery. This task remains a challenging open problem in the case of general metabolites. We recently solved this problem for a particular simpler case of peptidic metabolites (Mohimani et al, MS-DPR, J. Proteome Res. 2013). However, we can estimate the false discovery rate (FDR) of the database search in molDiscovery by generating a decoy database using the target-decoy approach (See subsection “Computing FDR” in Methods). The FDR corresponding to the score of each molecule-spectrum match is reported in both the online version of molDiscovery on GNPS and the command-line tool. We added a new paragraph to the Discussion section to discuss the confidence of the molecule-spectrum matches reported by molDiscovery as follows.

MolDiscovery computes the scores between molecules and mass spectra based on the log-likelihood ratio of a probabilistic model. The higher the score is, the more likely the spectrum is generated by the molecule than by random chance. In addition, molDiscovery estimates FDR using the target-decoy approach. While a Markov Chain Monte Carlo algorithm for computing p-value of molecule-spectrum matches in the case of peptidic molecules has been developed, extending this approach to general molecules is challenging as it involves generation of random molecular structures.

We also enlarged the subsection “Computing FDR” of the Methods section by adding an example of practical FDR application:

Supplementary Fig. 18 shows an example of N_{target} and N_{decoy} calculation at different score thresholds for a search of a Streptomyces dataset against AntiMarin.

Supplementary Figure 18: Number of identifications in the target and decoy databases at different score thresholds. We show the number of molecule-spectrum matches with **a** molDiscovery and **b** Dereplicator+ in the search of MSV00078604 *Streptomyces* dataset (~178,000 spectra) against the AntiMarin database (60,908 metabolites). The Y-axis is in log-scale. The number of hits in the decoy database gradually decreases with increasing score. A score threshold corresponding to zero hits in the decoy database separates the most reliable identifications (0% FDR). In this particular case, such score thresholds are 40 and 30 for molDiscovery and Dereplicator+, respectively. Note that these thresholds correspond to almost an order of magnitude more 0% FDR identifications by molDiscovery compared to Dereplicator+.

C1.7 for the output, in metabolism the software cannot discriminate between isobaric species (think of leucine/isoleucine) which often co-elute. For this, having a list of potential 'overlapping' species would be needed.

R1.7 In the case of isobaric species, we disagree with the reviewer. For example, in the case of leucine/isoleucine, since their chemical structures are different, the fragmentations predicted are also different. Thus, their matching scores to an experimental spectrum are usually different as well, and molDiscovery is capable of distinguishing them.

We added a new subsection “**Handling isobaric molecules**” to the Result section to explain the fragmentation of isobaric species as follows:

Handling isobaric molecules. Since isobaric molecules have different structures, their fragmentations generated by molDiscovery could be different (Supplementary Fig. 17). We searched the compounds from the GNPS spectral library that contain leucine or isoleucine, and created isobaric species in silico by changing leucines to isoleucines and vice versa. Among the three molecules, in two cases molDiscovery assigned a larger score to the correct molecule, while in the remaining case the scores assigned were nearly identical (Supplementary Table 10).

Supplementary Figure 17: Depth 1 fragmentations of **a** leucine and **b** isoleucine by molDiscovery. The shared masses of the fragments are in blue, while the different masses are in red.

GNPS ID	Original residue (score)	isobaric residue (score)
CCMSLIB00000854688	ile (108.067)	leu (105.927)
CCMSLIB00000855513	ile (36.7271)	leu (35.7906)
CCMSLIB00000848589	leu (15.3513)	ile (15.7267)

Supplementary Table 10: MolDiscovery scores for three compounds containing leucine or isoleucine and their corresponding isobaric species in the GNPS spectral library.

C1.8 Fig. 7: the third group from the left states "...non-mental compounds" I believe this is a typo?

R1.8 Yes, this is a typo thank you for catching it. We have updated Fig. 7 (now it is Fig. 6). See R3.24.

C1.9 can the tool be used to predict fragmentation of small molecules by diverse fragmentation modes? For instance ETD, CID, HCD,

R1.9 The probabilistic model of molDiscovery can be trained on spectra of different fragmentation modes and applied to corresponding types of spectra respectively. However, currently, we only provide a general version of molDiscovery that can be applied to any fragmentation modes, instead of separate pretrained models for different fragmentation modes for the following reasons. First, the majority of the training data of molDiscovery does not contain such fragmentation mode meta information. Second, on a subset of data that has such fragmentation mode information, we tested molDiscovery on the CID mode and HCD mode spectra. Supplementary Fig. 7 shows that the performance of molDiscovery does not change much after changing the fragmentation mode of the training data it uses.

Nevertheless, molDiscovery can be flexibly trained by the users using their own in-house spectra of a particular mode. In addition to the existing general pretrained model, we now have added the

pretrained models for CID and HCD spectra to our GitHub repository. We add the following sentences to the Discussion section:

Mass spectrometry fragmentation is influenced by experimental factors such as instrument type, dissociation technique, ionization mode, collision energy, etc. Currently, the default version of molDiscovery is trained on a combination of CID and HCD spectra. While molDiscovery also includes pretrained models consisting of CID only and HCD only spectra, our results show that these models do not outperform the default model on CID or HCD datasets. Moreover, while the default parameters are trained on positive mode spectra, molDiscovery still performs well on negatively charged spectra. Since molDiscovery supports training by custom spectral datasets, users can adapt the default model to their experimental settings by training on in-house data.

C1.10 can the tool deal with batch searches from multiple MSMS files?

R1.10 In the online version, users can select multiple MSMS files or a directory containing multiple MSMS files. In the command line version, users can specify the directory of input MSMS files, and molDiscovery will automatically search all the .mzML, .mzXML and .mgf files in the directory.

We have added instructions about how to use the online tool and highlighted the batch search feature in the Supplementary Note 2 Step 3, 3.1 and 3.2 (See R1.4). We also updated the documentation of the command-line version at <https://github.com/mohimanilab/molDiscovery> to show the batch search functionality of molDiscovery.

C1.11 similar to other prediction tools, the performance is affected by the molecular size of the target. Especially in the low m/z range, all tools seem to struggle (probably because less fragments are generated), unfortunately this is the among the most interesting range for metabolomics research. Can the authors comment on this?

R1.11 We have updated Fig. 7 (now it is Fig. 6, see R3.24 for details) and added the following sentence in the subsection “Performance on different compound classes and mass ranges” of Results to highlight this finding:

Similarly, molDiscovery performs equally or better than the competing methods in four out of six mass ranges (Fig. 6b). It is worth noting that in low mass ranges (<600 Da) the prediction accuracies of all the methods are lower than their performance in high mass ranges (>600 Da), which is probably due to the fact that large molecules tend to generate more fragment ions.

We also revised the following paragraph in the Discussion section:

MolDiscovery introduces an efficient algorithm to construct fragmentation graphs of small molecules, enabling processing molecules up to 2000 Da and molecular databases with large number of molecules. MolDiscovery is one of the most efficient methods in terms of both searching running time and memory consumption, which make it scalable to search of large-scale spectral datasets.

Furthermore, by training on the reference spectra from the GNPS spectral library, molDiscovery learned a probabilistic model that outperforms the other methods in overall accuracies, various mass ranges and compound classes. While molDiscovery is the best performing method in high mass range (>600 Da), similar to other database search methods, it has lower accuracy for molecules in low mass ranges (<600 Da). This is partially because fewer fragments are generated for molecules in this range. This makes it crucial to increase the accuracy of fragmentation models of small molecules.

Minor comments

N/A

Reviewer #2

C2.1 The current article focuses on a major challenge in molecular mass spectrometry – and moves the needle in this populated and venerable area. The work advances the implementation and scoring (including control of FDR for large datasets) of taking moderate and high resolution tandem mass spectrometric datasets and converting those into compound assignments from a database of tens of thousands of knowns. Further, the ability to assert detection of analogs of known compounds is also shown; this is akin to error-tolerant database searching in mass spec-based proteomics. Another application demonstrated is to detect natural products, with some examples of assignment to their BGCs. This is sophisticated algorithmic performance, and therefore, publication is recommended, with a few provisos provided in the following Major and Minor comments.

R2.1 Thank you for the positive words about our paper -- we have addressed your comments below. The changes are indicated in red in the revised manuscript and the Supplement.

Major comments

C2.2 “Our results show that molDiscovery greatly increases the accuracy of small molecule identification, while making the search an order of magnitude more efficient.”

Supporting evidence is shown in the manuscript; however this reviewer did not validate that the code base downloads, compiles and functions as advertised (e.g., generates the FDR estimation that shows better performance as shown in the manuscript).

R2.2 Thank you for your comment. Both the online version on GNPS and the command-line version of molDiscovery are available. We also added guidelines on how to use the online tool to Supplementary Note 2 (See R1.4). All the benchmarking results are also available at <https://github.com/mohimanilab/molDiscovery>.

C2.3 “Moreover, by applying molDiscovery on microbial datasets with both mass spectral and genomics data we successfully discovered the novel biosynthetic gene clusters of three families of small molecules.”

The Abstract-level claim needs to be tempered down. The 3 “novel” BGCs assigned to the compounds need to be carefully evaluated. These 3 examples are from known compound families and each one should be the result of a careful literature search to see if these assignments are indeed novel. In any case, they are “putative” assignments, and the results should be reported more carefully. In how many strains are these compounds observed? How are the masses related to those for known compounds in these molecular families?

Supplemental Figures S7 and S8 can be augmented to capture this information, and some if it should be reported in textually the Main Text. If indeed these assignments of known compounds (and their analogs) to their BGCs are all truly novel – that is a major accomplishment for the work.

R2.3 We have moved Supplementary Fig. 7 and 8 to the main text according to the reviewer comments and further polished all the three BGC Figures to show the post-modifications and the corresponding putative tailoring enzymes.

Figure 9: Putative BGC of dinghupeptin family in *Streptomyces* sp. NRRL B-5680. MolDiscovery identified four dinghupeptin variants at FDR 0%. After searching its genome using antiSMASH, we detected an NRP BGC with adenylation domains which show high specificity to the amino acids residues of dinghupeptin family molecules. We also used DFAST (Supplementary Table S5) and HMMsearch to annotate the genes in the gene cluster. The post-modifications and the corresponding putative tailoring enzymes are color coded.

Figure 10: Putative BGC of lipopeptin and neopeptin family in *S. hygroscopicus* NRRL B-1477. MolDiscovery identified two lipopeptin variants and neopeptin variants at FDR 0%. After searching its genome using antiSMASH, we detected a NRP BGC with adenylation domains which show high specificity to the amino acid residues of lipopeptin and neopeptin family molecules. We also used DFAST (Supplementary Table S6) and HMMsearch to annotate the genes in this BGC. The post-modifications and the corresponding putative tailoring enzymes are in the same color.

Figure 11: Putative BGC of longicatenamycin family in *S. rimosus* NRRL B-8076. MolDiscovery identified two longicatenamycin variants at FDR 0%. After searching its genome using antiSMASH, we detected an NRP BGC with adenylation domains which show high specificity to the amino acids residues of longicatenamycin. We also used DFAST and HMMsearch to annotate the genes in the gene cluster (Supplementary Table S7). The post-modifications and the corresponding putative tailoring enzymes are in the same color. It is known that hydroxylation (red) can be induced by the Taurine catabolism dioxygenase [50], and flavin reductase can involve in FAD-dependent hydroxylation [51]. 2-isopropylmalate synthase (blue) can function on the isopropyl group and elongate the chain (EC 2.3.3.13).

Moreover, we added Supplementary Tables 5-7 to show the detailed annotations of the genes in each of the BGC.

location	product	gene
58..732	TetR family transcriptional regulator	
complement(906..1148)	hypothetical protein	
complement(1767..2738)	sugar ABC transporter permease	
complement(2735..3817)	sugar ABC transporter permease	
complement(3814..5337)	sugar ABC transporter ATP-binding protein	
complement(5416..6399)	LacI family transcriptional regulator	
complement(7105..8094)	LacI family transcriptional regulator	
complement(8548..10116)	alpha-N-arabinofuranosidase	abfA_1
complement(10249..11811)	alpha-N-arabinofuranosidase	abfA_2
12286..14997	hypothetical protein	
complement(15176..16537)	hypothetical protein	
complement(17549..17746)	hypothetical protein	
complement(17746..18498)	hypothetical protein	
complement(18529..19776)	cytochrome P-450 like protein	
complement(19782..19997)	MbtH protein	mbtH
complement(20001..28664)	hypothetical protein	
28826..29341	hypothetical protein	
complement(30212..43642)	hypothetical protein	
complement(43747..46140)	hypothetical protein	
complement(46246..46617)	hypothetical protein	
complement(46920..48608)	hypothetical protein	
48820..50400	peptide ABC transporter substrate-binding protein	
50397..51383	peptide ABC transporter permease	
51386..53278	peptide ABC transporter ATP-binding protein	
53275..54078	hypothetical protein	
54075..55379	oxidoreductase	
complement(55369..56451)	hypothetical protein	
complement(56636..58819)	hypothetical protein	
59088..60437	glutamine synthetase	
60602..60793	hypothetical protein	
complement(61351..61728)	hypothetical protein	
complement(61980..62429)	cyclase	
complement(62736..63191)	hypothetical protein	

Supplementary Table 5: Gene annotation of dinghupeptin family BGC. The genes are annotated by DFAST.

location	product	gene
1..297	hypothetical protein	
501..1964	6-aminohexanoate-cyclic-dimer hydrolase	
complement(2177..3001)	DDE transposase	
complement(3138..3452)	hypothetical protein	
3583..3984	hypothetical protein	
4543..4998	hypothetical protein	
5019..5648	hypothetical protein	
complement(5972..6196)	hypothetical protein	
6286..6558	hypothetical protein	
6885..7265	hypothetical protein	
complement(8267..8671)	hypothetical protein	
complement(8709..9554)	hypothetical protein	
10227..11204	oxidoreductase	
complement(12255..13619)	MFS transporter	
complement(13629..15032)	hypothetical protein	
15240..16388	hypothetical protein	
16585..17538	"2,3-diaminopropionate biosynthesis protein SbnA"	cysM
17553..18572	"2,3-diaminopropionate biosynthesis protein SbnB"	ocd
18576..18785	hypothetical protein	
18782..19519	thioesterase	
19534..24468	hypothetical protein	
24564..27824	hypothetical protein	
27821..28621	hypothetical protein	
28746..29927	hypothetical protein	
complement(30021..31859)	SARP family transcriptional regulator	
complement(32071..33096)	2-oxobutyrate oxidase	
complement(33093..33782)	7-cyano-7-deazaguanine synthase	queC
complement(33784..34764)	hypothetical protein	
complement(34847..35437)	GTP cyclohydrolase 1	foiE
complement(35475..35753)	"6-carboxy-5,6,7,8-tetrahydropterin synthase"	
complement(35875..36588)	7-carboxy-7-deazaguanine synthase	queE
complement(36602..37384)	hypothetical protein	
complement(37758..40760)	SARP family transcriptional regulator	
complement(40780..40947)	hypothetical protein	
41800..42135	hypothetical protein	
complement(42775..43035)	hypothetical protein	
complement(43145..43909)	thiazole biosynthesis protein ThiJ	
43956..44909	AraC family transcriptional regulator	
45336..45962	resolvase	
46065..46439	hypothetical protein	
47147..47404	hypothetical protein	
47401..48699	MFS transporter	
48973..49656	hypothetical protein	
50297..50692	hypothetical protein	
51382..51702	hypothetical protein	
complement(52074..53123)	L-asparagine oxygenase	
complement(53230..53448)	protein mbtH	
complement(53519..60676)	hypothetical protein	
complement(60673..67518)	hypothetical protein	
complement(67577..83815)	hypothetical protein	
complement(84104..84379)	hypothetical protein	
complement(84487..84762)	hypothetical protein	
84866..85963	hypothetical protein	
85973..86224	hypothetical protein	
86352..87116	hypothetical protein	
complement(87317..87841)	integrase	

Supplementary Table 6: Gene annotation of lipopeptin family BGC. The genes are annotated by DFAST.

location	product	gene
3271..4461	amidase	amiB1
complement(4525..5523)	ABC transporter substrate-binding protein	
5693..6487	cobalamin/Fe3+-siderophore ABC transporter ATP-binding protein	
complement(6564..7784)	hypothetical protein	
8076..8843	hypothetical protein	
8871..9560	methyltransferase	
9541..9819	hypothetical protein	
complement(9836..10813)	hypothetical protein	
complement(10948..12174)	hypothetical protein	
12425..12667	hypothetical protein	
12805..13461	hypothetical protein	
complement(13535..14590)	hypothetical protein	
15272..16558	MFS transporter	
16776..17003	hypothetical protein	
17478..18545	prenyltransferase	
18542..19795	aminotransferase	
complement(19821..21287)	cytochrome P450	
complement(21284..21907)	ATP-binding protein	
complement(21888..22259)	hypothetical protein	
complement(22256..22678)	dynein regulation protein LC7	
complement(22697..24088)	ATPase	
complement(24655..25614)	hypothetical protein	
complement(25702..25896)	hypothetical protein	
complement(26542..26922)	hypothetical protein	
27175..28071	transcriptional regulator	
28263..28982	ABC transporter ATP-binding protein	
complement(29064..29714)	hypothetical protein	
30045..30314	hypothetical protein	
complement(30305..30679)	hypothetical protein	
31032..32432	3-isopropylmalate dehydratase large subunit	leuC
32447..33073	3-isopropylmalate dehydratase small subunit	leuD
33073..34284	aminotransferase	
34281..35309	protein AmbC	ambC
35306..36574	MFS transporter	
complement(36626..37270)	DNA-binding response regulator	
complement(37258..38514)	histidine kinase	
38699..39613	ABC transporter	
39618..42188	ABC transporter permease	
42900..44093	serine hydrolase	
44203..50637	hypothetical protein	
50641..63150	hypothetical protein	
63188..67786	hypothetical protein	
67842..68594	thioesterase	
68962..70464	tryptophan halogenase	
70502..71761	hypothetical protein	
71758..72279	FMN reductase	
72332..73159	amidinotransferase	
73203..74879	2-isopropylmalate synthase	leuA
74962..75777	indole-3-glycerol phosphate synthase 1	trpC1
complement(75749..76807)	anthranilate phosphoribosyltransferase 1	trpD1
77032..78459	GntR family transcriptional regulator	
complement(78468..79040)	glutamine amidotransferase	
complement(79037..80647)	hypothetical protein	
80857..82200	phospho-2-dehydro-3-deoxyheptonate aldolase	aroH
82399..83418	3-oxoacyl-ACP synthase	fabH
83448..85319	hypothetical protein	
85433..90103	hypothetical protein	
90100..98496	hypothetical protein	

Supplementary Table 7: Gene annotation of Ionicatenamycin family BGC. The genes are annotated by DFAST.

We also tempered down the claim in the abstract by replacing “novel biosynthetic gene cluster” with “putative biosynthetic gene cluster”:

Moreover, by applying molDiscovery on microbial datasets with both mass spectral and genomics data we successfully discovered the putative biosynthetic gene clusters of three families of small molecules.

We further include the BLAST search results of the three BGCs in the NCBI Nucleotide collection database and the MIBiG database and revised the description of the novel discoveries by clarifying the strains that each molecule was identified in as follows.

Moreover, molDiscovery successfully discovered novel BGCs for three small molecule families from *Streptomyces* dataset MSV000083738^{43,44}. MolDiscovery search results for this dataset are available at <https://github.com/mohimanilab/molDiscovery>. MolDiscovery identified dinghupeptins A-D⁴⁵ in multiple *Streptomyces* strains, including *Streptomyces* sp. NRRL B-5680, *S. varsoviensis* NRRL B-3589, *Ampullariella violaceochromogenes* NRRL B-16710 and *S. californicus* NRRL B-3320. AntiSMASH⁴⁶ revealed a non-ribosomal peptide (NRP) biosynthetic gene cluster in the genome of *Streptomyces* sp. NRRL B-5680, with seven adenylation domains that are highly specific to the amino acid residues of dinghupeptin molecules⁴⁷ (Fig. 9). After a BLAST search of this BGC against the NCBI nucleotide collection database⁴⁸ and the MIBiG database, the maximum coverages are 37% and 20% respectively, indicating this NRP BGC is not similar to any known BGC.

MolDiscovery detected lipopeptin A-C in multiple *Streptomyces* strains, including *S. hygrosopicus* NRRL B-1477, *S. rimosus* NRRL WC3874, *S. rimosus* NRRL WC3904 and *S. rimosus* NRRL B-2659. Moreover, neopeptin A-C are identified in *S. hygrosopicus* NRRL B-1477, *S. rimosus* NRRL WC387, *Streptomyces* sp. NRRL S-1824 and *Streptomyces* sp. NRRL WC3549. Genome mining revealed an NRP BGC in *S. hygrosopicus* NRRL B-1477 with seven adenylation domains, which are highly specific to the residues of neopeptin and lipopeptin (Fig. 10). It has been reported that these two families are structurally similar⁴⁹. Thus, this BGC could be responsible for the production of both families. BLAST results show the maximum coverages of this BGC by known sequences are only 25% and 15% in the NCBI nucleotide collection database and the MIBiG database.

In addition, molDiscovery identified longicatenamycin family of cyclic hexapeptides in *S. rimosus* NRRL WC3558, *S. rimosus* NRRL WC3925, *S. rimosus* NRRL B-8076, *S. lavendulae* NRRL B-2775, *S. rimosus* NRRL B-2661, *S. griseus* subsp. *rhodochrous* NRRL B-2931 and *S. rimosus* NRRL WC-3930. Genome mining of the *S. rimosus* NRRL B-8076 strain revealed a BGC highly specific to longicatenamycin (Fig. 11). Dereplicator+ and CSI-FingerID failed to discover longicatenamycin at 1% FDR threshold. BLAST results show neither the NCBI nucleotide collection database nor the MIBiG database have known genes homologous to this BGC.

Minor comments

C2.4 Suggest authors consider putting ~1-3 Figures into the Supplemental Information, e.g., for algorithm validation.

R2.4 We have moved Figures 3, 5, and 6 to the Supplementary Material.

C2.5 Discussion: “MolDiscovery is shown to be robust against dissociation technique” . Suggest rewording to be clearer to more general readers. Also, there are many dissociation techniques in mass spec., so this statement should be better justified/curtailed.

R2.5 We have revised the Results subsection “Evaluating molDiscovery sensitivity to platform variation” as follows.

It was shown that using training data from different dissociation techniques, e.g. CID and HCD did not affect performance of molDiscovery (Supplementary Fig. 7).

Reviewer #3

C3.1 The current paper discusses a software that can help in compound identification. The annotation of small molecules in complex matrices is an ongoing challenge, better algorithms are urgently needed. The MolDiscovery software described here is available as source code and also online via the GNPS database in a workflow format. The algorithms is very fast and has a small memory footprint, which is exciting.

The tool does not consider negative ionization mode spectra, which are important in metabolomics. This could be a future improvement. Many of the showcases are peptides or peptide-analogs and are very important but do not represent the full natural product space. Based on the limited and small benchmark set the claim that the tool works well for “small molecule fragmentations” has to be debated unless the authors show further proof. This proof could be easily done by including benchmarks from larger spectral sets including NIST or MassBank. The pseudomonas application is interesting, but not really convincing, here wider coverage should be applied including several mammalian, plant or even marine algal samples.

R3.1 Thank you for the suggestions on our paper – we have provided a point-by-point response to each of your comments below. The changes are indicated in red in the revised manuscript and the Supplement.

Major Comments

C3.2 Abstract: “The existing approaches use the domain knowledge from chemistry to predict fragmentation of molecules.” That is actually not true. See QCEIMS by Grimme 2013 which is a quantum chemical process to create fragmentations of small molecules (70 eV).

R3.2 We have revised this sentence as follows.

The existing approaches use chemistry domain knowledge or quantum simulation to predict fragmentation of molecules.

In addition, we add a sentence to briefly review QCEIMS in the introduction. See R3.3 for details.

C3.3 “The majority of in silico database search methods are rule-based models that incorporate domain knowledge such as bond types, hydrogen rearrangement, and dissociation energy [16, 17,

18,19, 20, 21] to predict fragmentations of small molecules and score small molecule-spectrum pairs. However, due to the complex rules involved in small molecule fragmentation, these methods are computationally insufficient for heavy small molecules. Moreover, these predictions, which are based on expert knowledge, fail to explain many of the peaks in mass spectra.”

a. Problem-1: it would be worthwhile to name the most successful approaches (MetFrag, CFM-ID, CSI-FingerID, MS-Finder, ChemDistiller). Many of these tools have been successfully used in CASMI (small molecule) contests or had very elaborate validations on thousands of small molecules (ChemDistiller, CFM-ID, CSI-FingerID). So simple citations are not sufficient to explain why they are discarded.

R3.3 We have extended this paragraph to name and review the most successful approaches as follows.

The majority of *in silico* database search methods are rule-based models that incorporate domain knowledge to score small molecule-spectrum pairs. EPIC uses heuristic penalties for matching fragment ions to score molecule-spectrum pairs¹⁵. MAGMa+ further scores for both matched and missed peaks¹⁶. MassFrontier is a commercial software that utilizes a large number of fragmentation mechanism rules to predict fragmentation spectra¹⁷. MetFrag is based on weighted peak count and bond dissociation energy to compute matching scores¹⁸. MIDAS takes account of fragment-peak matching errors¹⁹. MSFinder introduces the hydrogen rearrangement rules for fragmentation and scoring²⁰. Unlike the other methods, QCEIMS uses quantum chemical simulation to predict mass spectra²¹. However, due to the limitation of the rules and heuristic parameters, these methods often fail to explain many of the peaks in mass spectra.

Recently, spectral libraries with tens of thousands of annotated mass spectra of small molecules have emerged, paving the path for developing machine learning based methods to improve sensitivity and specificity of *in silico* database search. CFM-ID applies stochastic Markov process to predict fragmentation spectra²². CSI:FingerID predicts a molecular fingerprint based on mass spectra and searches the fingerprint in a molecular database²³. ChemDistiller combines both molecular fingerprint and fragmentation information to score the molecule spectrum matches²⁴. However, the existing methods do not perform well for super small molecules (less than 400 Da), and are computationally insufficient for heavy small molecules (greater than 1000 Da).

C3.4 Problem-2: It would be useful to define “heavy” with a specific molecular weight and then also proof this assertion in the manuscript. Many of the tools can handle molecules up to 1500 Dalton.

R3.4 We have now defined the heavy small molecule (greater than 1000 Da) in the introduction (See R3.3). In addition, we further compare the running time of different methods to show that molDiscovery is the most efficient one. We added the following paragraph to the subsection “Benchmarking *in silico* database search on spectral libraries.” of the Results section:

We also evaluated the running time based on the mass ranges of the correct molecule matches (Fig. 4). We found that the highly accurate methods, CFM-ID and CSI:FingerID, have overall running times that scale poorly with molecular weight. For spectra of molecules with masses >1000 Da we found that CSI:FingerID and CFM-ID take on average 5 hours, 13 minutes, 10

seconds and 1 hour, 55 minutes, 34 seconds, respectively. Meanwhile, molDiscovery takes only 6 minutes and 24 seconds on average for the same mass range.

Figure 4: Running time comparison. (A) Efficiency of different methods for compounds in the GNPS subset within different mass ranges. Inputs that caused any method to crash or exceed 20 GB of memory usage were excluded from this analysis. All reported running times and memory usages are without preprocessing (molDiscovery, Dereplicator+ and CFM-ID could be run with or without preprocessing).

Based on this running time of the search of the GNPS spectral library against DNP, we further estimate the running time of all the methods for searching large-scale spectral datasets (8 million spectra of 46 GNPS datasets) against AllDB. Except for molDiscovery, all the methods will take years to finish searching. We added the following paragraph to the subsection “Searching large-scale spectral datasets” of the Results section:

MolDiscovery search took 34 days on 10 threads, which is very close to the projected 329 days on a single thread. CSI:FingerID, CFM-ID, Metfrag and MAGMa+ and are not benchmarked, as the searches would have taken years (see projections in Supplementary Fig. 15). It is worth noting that when searching such large-scale spectral datasets, methods that require preprocessing (molDiscovery and CFM-ID) are much more efficient than others, as they only need to preprocess the molecule database once, which can be used for searching any future spectra efficiently. The long running times of CSI:FingerID and MetFrag are probably due to the inefficient combinatorial algorithms.

Supplementary Figure 15: Projected running time (preprocessing and search combined) of each method on the combined 46 GNPS spectral datasets listed in Supplementary Table 1 against AllDB. Projection assumes the running time of each method scale with $O(mn)$ where m is the number of compounds in the chemical database and n is the number of spectra. Preprocessing running times are assumed to scale linearly with the number of compounds in the chemical database. It is estimated that it will take molDiscovery 329 days (4 days for preprocessing and 325 days for searching) to finish running, while it will cost CFM-ID 2,068 days (1,402 for preprocessing and 666 day for searching), Dereplicator+ 2997 days (without preprocessing), MAGMA+ 45,988 days, MetFrag 199,139 days, and CSI:FingerID 473,164 days for running.

We also showed the projected run time of all the methods for searching the MoNA subset against a large molecular database (about 1.3 million in bioactive-PubChem). Again, molDiscovery performed best in terms of both running time and accuracy (See R3.19).

C3.5 c. Problem-3: In many cases the assertion that they rely on “expert knowledge” or use only rule-based approaches is not true, CFM-ID, SF-Matching and CSI-FingerID and others are machine learning based, no expert knowledge required.

R3.5 We have revised the introduction to review methods that are only based on rules and heuristic parameters and those that are based on machine learning separately. Please see our response R3.3 for details.

C3.6 The authors claim the algorithm can be used for identifying metabolite / small molecules. However, most Figures shown in the paper are about large molecule (>800-1000 Da) and mostly peptides. Most metabolites / small molecules have very low molecular weight. For example, 96.5% of molecules in the KEGG database have mass <1000 Da (<https://doi.org/10.1093/bioinformatics/btn603>). In the NIST Tandem Mass Spectral Library 2020 release, more than 98% of spectra have a molecular weight less than 1,000 Da, more than 93% spectra have molecular weight less than 600 Da. To have a general interest in metabolite /

small molecules identification, more data/examples about analyzing small molecules (<600 Da) should be presented.

R3.6 One of the most important applications of metabolomics is to identify secondary metabolites, which contain both heavy small molecules (fatty acids, non-ribosomal peptides, etc.) and light small molecules (alkaloids, terpenes, polyketides, etc.). These secondary metabolites are widely used as anti-viral, anti-bacterial and anti-tumor drug leads.

In this paper, we show that molDiscovery can successfully identify both the heavy and light small molecules. In Table 2, 9/19 molecules are less than 600 Da. We have added a column “Mass (Da)” to this table as follows:

molecule	Type	Mass (Da)	MiBiG ID	Organism	GenBank ID	taxa similarity	MSV ID	Blast Coverage	FDR (%)	Score
xantholysin A	NRP	1775.08	BGC0000463	Pseudomonas mosselii BW11M1	GCA_001562525.1	species	MSV000085018	1	0	171.16
xantholysin B	NRP	1761.07	BGC0000463	Pseudomonas mosselii BW11M1	GCA_001562525.1	species	MSV000085018	1	0	141.45
Orfamide C	NRP	1266.81	BGC0000399	Pseudomonas protegens Pf-5	GCA_000012265.1	strain	MSV000085018	1	0	130.09
xantholysin C	NRP	1801.1	BGC0000463	Pseudomonas mosselii BW11M1	GCA_001562525.1	species	MSV000085018	1	0	121.87
Orfamide A	NRP	1294.84	BGC0000399	Pseudomonas protegens Pf-5	GCA_000012265.1	strain	MSV000085018	1	0	116.64
hygromycin A	Saccharide	511.169	BGC0000698	Streptomyces hygroscopicus subsp. hygroscopicus NRRL B-1477	GCA_000721535.1	species	MSV000083738	0.95	0	76.3
Orfamide B	NRP	1280.82	BGC0000399	Pseudomonas protegens Pf-5	GCA_000012265.1	strain	MSV000085018	1	0	75.54
chlortetracycline	PK	478.114	BGC0000209	Streptomyces sp. NRRL B-5680	GCA_000719415.1	family	MSV000083738	0.96	0	64.05
oxytetracycline	PK	460.148	BGC0000254	Streptomyces rimosus subspecies rimosus NRRL B-2626	GCA_000721045.1	species	MSV000083738	1	0	60.11
PK-1	PK/NRP	342.252	BGC0000972	Escherichia coli IHE3034	GCA_000025745.1	species	MSV000078995	1	0	51.13
nostopeptolide A	PK/NRP	1066.57	BGC0001028	Nostoc punctiforme PCC 73102	GCA_000020025.1	genus	MSV000078891	0.98	0	50.72
PK-2	PK/NRP	439.341	BGC0000972	Escherichia coli IHE3034	GCA_000025745.1	species	MSV000078995	1	0	49.17
PK-3	PK/NRP	340.236	BGC0000972	Escherichia coli IHE3034	GCA_000025745.1	species	MSV000078995	1	0	48.03
pilpastatin	NRP	1462.8	BGC0000407	Bacillus subtilis subsp. subtilis 168	GCA_000009045.1	species	MSV000084117	1	0	43.44
CDA4a	NRP	1494.52	BGC0000315	Streptomyces coelicolor A3(2)	GCA_000203835.1	strain	MSV000078839	1	0	43.24
Xenocoumacin 1	PK/NRP	465.259	BGC0001054	Xenorhabdus miraniensis DSM 17902	GCA_002632615.1	genus	MSV000081063	0.9	0	41.47
candicidin	PK	1108.57	BGC0000034	Streptomyces sp. CNY228	GCA_000377545.1	genus	MSV000078839	1	0.01	32.43
yersiniabactin	NRP	481.116	BGC0000467	Escherichia coli MS 21-1	GCA_000164355.1	order	MSV000082045	1	0.01	31.12
sangivamycin	other	309.107	BGC0000879	Streptomyces rimosus NRRL WC3904	GCA_000720725.1	species	MSV000083738	0.95	0.01	26.96

Table 2: MolDiscovery identified 19 molecules in bacterial isolates that contained their known BGCs. A genome is reported to contain a known BGC from the MiBiG database³⁵ if the genome has total BLAST hit⁴³ to the BGC with at least 90% coverage and 70% identity. Taxa similarity means similarity between the taxon of source organism and the organism reported in MiBiG. PK stands for polyketides, PK-1 stands for N-myristoyl-D-asparagine, PK-2 stands for (R)-N1-((S)-5-oxohexan-2-yl)-2-tetradecanamidosuccinamide, and PK-3 stands for cis-7-tetradecenoyl-D-asparagine.

Moreover, Fig. 6b showed that molDiscovery had comparative accuracies with the other methods for molecules less than 600 Da (See R3.24). We have updated the subsection “Performance on different compound classes and mass ranges” and the Discussion section accordingly (See R1.11).

C3.7 “MolDiscovery identified 56,971 small molecules at 0% false discovery rate (FDR),”

Please add to supplement with reference, links and IDs. Make sure to create histograms with the most annotated compounds. It could be 10,000-times alanine, 10,000-times glucose etc. So the number 56,971 does not say anything about the performance. Also in this case please match against the latest small molecule databases such as NIST20 or Massbank. Compare the annotation rates for spectral analysis.

R3.7 We have added a link to GitHub with the list of all and unique compounds to the paper (<https://github.com/mohimanilab/molDiscovery>). In Fig. 7b, we also showed the number of unique compounds identified by molDiscovery. We have revised the sentence in the introduction as follows.

molDiscovery identified 3,185 unique small molecules at 0% false discovery rate (FDR), a six times increase compared to existing methods.

We also added the spectral library search results of NIST17 to the subsection “Searching large-scale spectral datasets” of the Results section:

Searching large-scale spectral datasets. We benchmarked the performance of molDiscovery, Dereplicator+ and spectral library search on 46 GNPS datasets containing a total of 8 million tandem mass spectra (Supplementary Table 1). In contrast to the GNPS spectral library where all the spectra are annotated and their molecules are known, the spectra in the GNPS datasets are not annotated, and they could correspond to novel or known molecules. These spectra are searched against NIST17 by spectral library search and against AllDB by molDiscovery and Dereplicator+.

We compared the annotation rate (Supplementary Table 1), number of unique compounds (Fig. 7) and most annotated compounds (Supplementary Table 2) between spectral library search baseline and molDiscovery. Our results show that as expected, molDiscovery can annotate more tandem mass spectra than spectral library search. The top 10 compounds identified by molDiscovery at different mass ranges are shown in Supplementary Table 3.

MassiveID	No. spectra	molDiscovery	spectral library
MSV000078839	403604	0.273	0.002
MSV000078995	818	0.549	0.0
MSV000082831	36566	0.216	0.0
MSV000085214	3284	0.011	0.0
MSV000085180	9095	0.457	0.0
MSV000085023	2372	0.113	0.0
MSV000078836	481548	0.224	0.007
MSV000085003	8526	0.073	0.0
MSV000085123	769	0.0	0.027
MSV000079015	6085	0.508	0.009
MSV000083734	289693	0.886	0.0
MSV000084954	15513	0.819	0.021
MSV000084884	85130	0.287	0.002
MSV000081063	28563	0.751	0.019
MSV000084117	12287	0.392	0.0
MSV000084475	638641	0.491	0.0
MSV000079139	4435	0.003	0.028
MSV000084771	442	0.656	0.0
MSV000085027	426	0.263	0.021
MSV000084945	1550594	0.192	0.015
MSV000081318	6376	0.362	0.0
MSV000085159	188737	0.177	0.0
MSV000080251	1462003	0.433	0.006
MSV000079284	27818	0.041	0.0
MSV000084723	136043	0.065	0.0
MSV000085192	1634	0.234	0.0
MSV000083648	7	0.857	0.0
MSV000081504	607	0.565	0.0
MSV000078891	207413	0.369	0.0
MSV000078847	28615	0.233	0.001
MSV000085158	25606	0.381	0.0
MSV000084674	28854	0.484	0.046
MSV000082285	943	0.525	0.008
MSV000083295	3850	0.227	0.002
MSV000083081	1389	0.672	0.0
MSV000079519	49352	0.315	0.051
MSV000078556	61970	0.123	0.002
MSV000083738	409245	0.531	0.0
MSV000085179	7853	0.012	0.0
MSV000084989	808	0.037	0.0
MSV000080427	12102	0.038	0.0
MSV000082045	1665897	0.405	0.0
MSV000085018	44223	0.933	0.0
MSV000078850	59175	0.317	0.002
MSV000085026	1872	0.014	0.0
MSV000085032	2660	0.072	0.0

Supplementary Table 1: 46 GNPS spectral datasets analyzed in the paper and their annotation rates by molDiscovery and spectral library search. There are 8,013,433 spectra in total. NIST17 spectral library is searched with cosine similarity threshold 0.7, while molDiscovery score threshold is 7 ($\sim 1\%FDR$)

spectral library search		molDiscovery	
compound	#spectral identifications	compound	#spectral identifications
L-Tryptophan	25648	Fenestin_A	67253
Cholic acid	21412	Cyclo(leucylprolyl)-(3R,8aS)-form	41371
Phe-Pro	16856	6-(1-Hydroxy-1-methylethyl)-3-(2-methylp	22781
Trimethoprim	11057	2-Chloro-6-Methyl-Aniline	6354
Compound A	10636	Glycylprolylphenylalanylprolylisoleucine	6156
13-Docosamide, (Z)-	9157	5-(2-Hydroxy-2-methylpropyl)-3-isopropyl	5991
Ile-Pro-Ile	7978	PE(16:0/17:1(9Z))	5882
DL-Indole-3-lactic acid	7512	cyclo-[Phenylalanyl-prolyl]2	5555
Bis(2-ethylhexyl) phthalate	7239	Nummularine_K_Dihydro	5546
Dibutyl phthalate	6797	cyclo-(propyl-propyl-tyrosyl-tyrosine)	5373

Supplementary Table 2: Most annotated compounds in spectral library search and molDiscovery search in 46 GNPS datasets. The similarity cosine score threshold of spectral library search against NIST17 is 0.7. The FDR threshold of molDiscovery search against AllDB is 1%. Compound A stands for 1,2-Di-(9Z-octadecenyl)-sn-glycero-3-phosphoethanolamine.

Figure 7: Number of unique discoveries by **a** spectral library search against NIST17 and **b** in silico database search against AllDB.

SpecFile	Name	Score	Mass	MZ	RT	Adduct
< 200						
MSV000085018	Chokol A	40.0053	198.162	199.169	1346.53	M+H
MSV000085018	"3,4-Didehydro-beta-ionol"	39.8983	192.151	193.159	878.585	M+H
MSV000085018	4-n-Hexylphenol	39.6969	178.136	179.143	1013.29	M+H
MSV000084674	"Kobifuranone B 2-Deoxo, 5-oxo, 3,4-dihydro"	39.293	196.11	197.117	145.887	M+H
MSV000085158	5-(but-3-en-1-yl)-3-propylfuran-2(5H)-one	38.8054	180.115	181.122	735.623	M+H
MSV000085018	"5(13),7-Megastigmadien-9-one -(E)-form"	38.5856	192.151	193.159	882.017	M+H
MSV000078891	Compound_196.11	38.2996	196.11	197.117	648.557	M+H
MSV000084674	Pestalafuranone G	37.7613	196.11	197.117	150.064	M+H
MSV000085018	"8,10-Pentadecadiene-2,4,6-triyne"	37.4299	196.125	197.132	1321.34	M+H
MSV000084945	"11,12,13-Trinor-2,6-farnesadiene-1,10-diol"	36.6533	198.162	199.17	696.417	M+H
200-400						
MSV000085018	"8Z,11Z,14Z,17Z,20Z,23Z-hexacosahexaenoic acid"	116.156	384.303	385.31	1302.12	M+H
MSV000084475	Raspailol A Raspailol A	110.732	374.282	375.29	353.863	M+H
MSV000084475	3alpha-Hydroxy-5beta-chole-7-en-24-oic Acid	110.701	374.282	375.29	355.213	M+H
MSV000084475	"3-Hydroxychole-11-en-24-oic acid (3 α ,5 β) - form"	109.454	374.282	375.29	355.012	M+H
MSV000085018	"25-Dehydrovitamin D3 (5Z,7E)-(3S)-9,10"	107.401	382.324	383.331	1305.55	M+H
MSV000084475	ethyl 10-hydroxy-docosapentaenoate	107.135	374.282	375.29	354.87	M+H
MSV000083738	vanchrobactin	106.099	397.16	398.167	446.19	M+H
MSV000084475	cholacalcioic acid	105.875	372.266	373.274	289.133	M+H
MSV000083738	Ficus Latex peptide 3	105.866	386.264	387.271	1263.63	M+H
MSV000084475	3b-Hydroxy-5-cholenoic acid	105.516	374.282	375.29	353.81	M+H
400-600						
MSV000083738	Antibiotic LL-BM 547alpha	154.272	557.231	558.239	178.786	M+H
MSV000084954	Torularhodin 16'-Alcohol	148.223	550.417	551.423	312.446	M+H
MSV000083738	Heterobactin A	148.005	598.202	599.211	2154.92	M+H
MSV000084954	Alloxanthin Alloxanthin	146.611	564.397	565.401	283.176	M+H
MSV000084954	"(E)-3',4'-didehydro- β , ψ -caroten-16'-ol"	145.624	550.417	551.423	311.062	M+H
MSV000084954	Crocoxanthin	142.812	550.417	551.423	310.387	M+H
MSV000084954	Monodoxanthin 3'-Deoxy	141.02	550.417	551.423	311.472	M+H
MSV000084954	Retrodihydro-g-carotene	139.89	534.423	535.427	313.449	M+H
MSV000083738	13-Hydroxyglucopiericidin A	137.196	593.32	594.327	3300.44	M+H
MSV000083738	"24-Hydroxydammara-20,25-dien-3-one"	135.903	440.365	441.372	2914.57	M+H
600-800						
MSV000084945	Ornibactin C4 N5-Deacyl	212.54	736.397	737.404	176.698	M+H
MSV000081063	Taxillaid C	195.483	793.531	794.533	612.231	M+H
MSV000083738	Antrimycin D	191.778	727.386	728.394	1949.09	M+H
MSV000084945	a-Substance Ib	184.139	685.391	686.399	247.877	M+H
MSV000081063	Ambactin	183.167	750.406	751.409	395.928	M+H
MSV000081063	Xenobovid B	182.022	793.531	794.535	608.085	M+H
MSV000083738	Monamycin-B3	173.248	677.411	678.417	3688.46	M+H
MSV000083738	Lonicatenamycin	173.163	776.362	777.371	2858.14	M+H
MSV000084884	Alterochromide A	167.987	751.354	752.355	203.68	M+H
MSV000085018	haprolid	167.732	682.431	683.441	2503.76	M+H
800-1000						
MSV000081063	Taxillaid A	209.695	807.547	808.549	627.514	M+H
MSV000083738	Surugamide G	206.771	883.59	884.594	2615.94	M+H
MSV000079519	E'Champacyclin'	202.923	897.605	898.611	286.793	M+H
MSV000083738	"TG(16:0/20:1(11Z)/18:2(9Z,12Z))"	184.924	882.768	883.776	3829.81	M+H
MSV000078891	Anabaenopeptin NZ857	183.617	857.432	858.435	624.713	M+H
MSV000081063	Xentrivalpeptide A	180.643	859.484	860.487	558.542	M+H
MSV000081063	Szentiamide	171.439	837.406	838.41	509.547	M+H
MSV000079519	Ogipeptin A	168.914	954.597	955.597	269.902	M+H
MSV000081063	Xenobovid C	160.835	821.563	822.564	662.551	M+H
MSV000080251	largamide A methyl ester	158.361	855.438	856.443	200.125	M+H
1000 \geq						
MSV000085018	massetolide J	245.834	1111.67	1112.68	1536.35	M+H
MSV000085018	viscosin	221.532	1125.69	1126.69	1505.12	M+H
MSV000084117	Esperin	206.03	1035.68	1036.69	342.024	M+H
MSV000083738	Actinomycin monolactone	205.893	1272.64	1273.65	3499.08	M+H
MSV000085018	Gacamide A	197.487	1393.84	1394.84	1528.63	M+H
MSV000085192	Sameuramide	192.685	1015.55	1016.55	255.397	M+H
MSV000085018	Antibiotic MA 026 Antibiotic MA 026	189.546	1775.08	888.547	1789.11	M+2H
MSV000085018	Massetolide F	188.566	1125.69	563.852	1612.54	M+2H
MSV000085192	Compound_1001.53	187.785	1001.53	1002.54	260.242	M+H
MSV000083738	callipeltin B	186.724	1030.54	1031.54	3525.33	M+H

Supplementary Table 3: Top identifications of molDiscovery in the 46 datasets in different mass ranges. Compound_196.11 represents 3-(1-Hydroxyhexyl)-5-methylene-2(5H)-furanone. Compound_1001.53 represents "3-acetamido-22-benzyl-10-<1<(3-hydroxy-4-methyl-2-propionamidopentanoyl)oxy-2-methylpropyl>-4-isopropyl-7-(1-methoxyethyl)-19-methylene-8,13,14,16,20-pentamethyl-1,5-dioxo-8,11,14,17,20-pentaazacyclodocosane-2,6,9,12,15,18,21-heptone"

C3.8 Dataset section: "The GNPS spectral library contains 8,964 small molecule-spectrum matches"

Looking at the GNPS download link (<https://gnps-external.ucsd.edu/gnpslibrary>) That is actually not true. There are 16,610 records with unique SMILES and 8,374 records with InchiKeys. The intrinsic problem here is that out of 156,355 GNPS records (ALL_GNPS.msp) very few actually have annotated structures.

R3.8 Since GNPS is a community-curated database, the size is increasingly growing. Moreover, many of the spectra in GNPS are duplicates. We initially used a subset of GNPS from 2017 for benchmarking. We have updated to a version downloaded on October 18, 2020. We have also added the following paragraph to the subsection “Datasets” to clarify the exact subset of GNPS that we used:

Datasets. The GNPS spectral library and MassBank of North America (MoNA) were used as training and testing data. For the GNPS spectral library, we selected molecule-spectrum pairs from the NIH Natural Products Library in GNPS (Oct 18, 2020) with unique spectrum (SPLASH values [26], please see Supplementary Fig. 1), unique molecule (first 14 characters of InChIKey) and precursor m/z that could be explained within an absolute error tolerance of 0.02 Da. If a molecule corresponded to multiple spectra, a random spectrum was selected. This left us with 4,437 and 3,964 spectra in positive and negative mode, respectively. With similar filtering steps as above and only keeping spectra with at least 5 peaks from Vaniya/Fiehn Natural Products Library obtained on Q-Exactive HF instruments (VF NPL QEHF), 6,528 singly-charged and 163 doubly-charged spectra in positive mode were selected. We also selected 2,382 singly-charged NIST20 spectra from Q-TOF instruments using the same filtering steps.

C3.9 The MONA positive mode LC-MS/MS libraries has 91,774 unique records based on around unique 13,600 structures, but for a single given molecule, multiple MS/MS spectra with different collision energies or even based on different instruments or in-source voltages could be derived. So simply filtering by unique structure is not sufficient.

This ignores the multiple CID or HCD voltages which can lead to very different spectra. CFM-ID for example was trained on different CID voltages and correctly takes this into account. So here I recommend to retrain based on all MassBank, NIST or MONA spectra. If retraining is not required include extended validation.

R3.9 We tested molDiscovery on CID mode and HCD mode spectra. Supplementary Fig. 7 shows that the performance of molDiscovery does not change much after changing the fragmentation mode of the training data it uses. Due to the lack of metadata of different voltages, we have only provided separate models for CID, HCD-high, and HCD-low on GitHub. Nevertheless, molDiscovery can be flexibly trained by the users using their own in-house spectra of a particular mode. See R1.9 for details.

C3.10 The paper utilizes Dereplicator or Dereplicator+ as baseline, however some of the best algorithms for small molecule annotations, including MetFrag or ChemDistiller are not even included or benchmarked here, (only CSI-FingerID and CFM-ID). These tools were comprehensively validated in their publications and also participated in the CASMI contests. Dereplicator+ has not been evaluated in such a way or even benchmarked against the best tools available, except in this paper (or the dereplicator publication). I recommend to utilize at least

one more (ChemDistiller or Metfrag) and perform a benchmark test on 10,000 NIST or MONA database structures. Filtering of compounds and spectra that were used during training will be difficult, but this benchmark set should give a good baseline.

R3.10 We have added benchmarks against MetFrag on the updated GNPS spectral library against DNP. Please see R3.8 for a description of the updated GNPS subset used. Additionally, we have tested all the tools on 6,528 positive mode spectra (2,054 unique compounds) from the VF-NPL-QEHF sublibrary in MoNA. We searched these spectra against 10,124 unique compounds (selected by the first 14 characters of their InChIKeys) corresponding to all [M+H]⁺ spectra from the MoNA's LC-MS/MS Positive Mode dataset (93,889 spectra). Of these spectra, all but 342 corresponded to compounds that CSI:FingerID used in its training dataset.

Below are plots of top 1, 3, 5, and 10 accuracies for each method evaluated on both GNPS and MoNA. Supplementary Fig. 5 is the results on all the spectra described above, while Fig. 3 excludes the CSI:FingerID training data. Note that for molDiscovery we did not retrain the model on MoNA, and simply used the model trained on the GNPS dataset. The Figures show that molDiscovery is one of the best performing methods on either dataset. CSI:FingerID outperforms other methods a lot on the GNPS dataset including its training data, but the accuracy drops dramatically on the dataset excluding its training data.

Figure 3: Top $K = 1, 3, 5,$ and 10 accuracy for all tested methods when searching **a** 194 spectra from the GNPS spectral library against 77,057 molecules from DNP and **b** 342 spectra from MoNA against 10,124 molecules from MoNA (spectra from molecules overlapping with CSI:FingerID training data are removed). Ties in scores were evaluated by setting the rank of the candidate compounds with tied scores to the average of all of those ranks. See Supplementary Note S3 for detailed parameter settings of these methods.

Supplementary Figure 5: Top 1, 3, 5, and 10 accuracy for all tested methods when searching **a** 4,437 spectra from the GNPS spectral library against 77,057 molecules from DNP and **b** 6,528 spectra from MoNA against 10,124 compounds from MoNA. Ties in scores were evaluated by setting the rank of the candidate compounds with tied scores to the average of all of those ranks. Note that 95.6% of GNPS spectra and 94.8% of MoNA spectra came from compounds found in the CSI:FingerID training dataset.

We further explore the reasons for the large drop in CSI:FingerID accuracy. Based on comment C3.13 we used InChIKeys instead of canonical SMILES to match CSI:FingerID training structures to our GNPS dataset and found that, as the reviewer noted, filtering for unique structures based on canonical SMILES caused us to miss many matches between the GNPS dataset and the CSI:FingerID training dataset. This is the primary cause of the large drop in CSI:FingerID accuracy that we observe above. See also below a plot comparing the accuracy of molDiscovery and CSI:FingerID based on the Tanimoto similarity between the true compound and the training dataset for each method. Note that we see a large jump in accuracy for CSI:FingerID when we reach a Tanimoto similarity of 1.0, indicating CSI:FingerID bias towards the very similar structure it has observed.

Supplementary Figure 6: Database search accuracy according to the Tanimoto similarity between training compounds and testing compounds. Note that there is a large jump in accuracy for CSI:FingerID when we reach a Tanimoto similarity of 1.0, indicating that CSI:FingerID is biased towards the molecular structures similar to its training data.

Moreover, we updated benchmarks of the efficiency of all the methods. Below are the updated run time and memory usage table incorporating the addition of MetFrag, the updated GNPS subset (R3.8), and the fixed CFM-ID parameters (R3.12). Additionally, we have included preprocessing run time for CFM-ID and molDiscovery, the only two methods with a molecular preprocessing procedure more complex than simple format conversion. MolDiscovery is both the fastest and the most memory-efficient method.

Method	Searching time (d-h:m:s)	Searching peak memory (GB)	Preprocessing time (d-h:m:s)	Preprocessing peak memory (GB)
molDiscovery	00:29:04	0.15	10:40:25	16.38
CFM-ID	00:59:37	0.012	157-08:28:08	3.839
Dereplicator+	04:28:10	19.98	NA	NA
MAGMa+	2-20:34:37	0.51	NA	NA
MetFrag	12-08:57:24	4.01	NA	NA
CSI:FingerID	29-09:35:05	19.68	NA	NA

Table 1: Running time and peak memory usage of all methods in searching GNPS against DNP. All reported running times are on a single CPU. Note that CFM-ID running time and memory usage does not include filtering database candidates by precursor m/z. All running times are only for the spectra for which the method completed successfully (exit code 0). For CSI:FingerID 423 spectra either failed with a “feasibility” error or required more than 20GB of memory, and are not included in values in this table. Note that given a small molecule structure database, molDiscovery and CFM-ID only needs to preprocess the database for once. For searching very large spectral datasets, they are usually more efficient than other methods.

We revised the subsection “Benchmarking in silico database search on spectral libraries” of the Results section as follows:

Benchmarking in silico database search on spectral libraries. MolDiscovery is compared with five other state-of-the-art methods, including Dereplicator+, MAGMa+, CFM-ID, CSI:FingerID and MetFrag on the GNPS spectral library and MoNA. The database search results show that molDiscovery on average can correctly identify 43.3% and 64.3% of small molecules in the testing GNPS and MoNA data, respectively, as top-ranked identifications (Fig. 3).

MolDiscovery is the best performing method in the MoNA dataset and slightly worse than CFM-ID (43.8%) in the GNPS dataset. Note that CSI:FingerID's performance significantly drops when removing its training data (Supplementary Fig. 5). Our results indicate that CSI:FingerID is biased towards the structures that are very similar to its training data, while molDiscovery's performance does not depend on the similarity between training and testing data (Supplementary Fig. 6).

MolDiscovery is also the fastest and one of the most memory-efficient methods for searching GNPS against DNP. CFM-ID is slightly slower than molDiscovery in the searching stage. In the preprocessing stage, molDiscovery is over 300 times faster than CFM-ID (Table 1).

C3.11 The software does not discuss rearrangement reactions. However they are important in mass spectrometry and several tools including CFM-ID or MS-Finder integrate such rules. Rearrangements are very common in molecules with small molecular weight. How does the algorithm deal with this? This goes back to my prior point that the MolDiscovery software needs to be properly benchmarked against long-standing and existing tools.

R3.11 MolDiscovery associates the hydrogen rearrangement rules with the type of bond that is fragmented. We add/subtract the corresponding mass of the hydrogen to the fragment accordingly after every fragmentation. We have added a Supplementary Table 9 to explain the rearrangement rules used by molDiscovery.

Fragment A	Fragment B	Hydrogen rearrangement
C	C	A+H B-H or A-H B+H
C	N	A-H B+H
C	O	A-H B+H
C	S	A-H B+H
C	P	A+H B-H
O	P	A+H B-H

Supplementary Table 9: Rearrangement rules used in molDiscovery. MolDiscovery associates the hydrogen rearrangement rules with the type of bond that is fragmented. All the bonds in the table are single bonds. For example, whenever C-O bond is disconnected, a -H mass shift is considered for the fragment on the carbon side, and +H mass shift is considered for the fragment on the oxygen side. When a C-C bond is disconnected, both rearrangement scenarios are considered.

In the main text accordingly, we also added the following sentence to the subsection “Constructing fragmentation graphs of small molecules.” of the Methods section as follows:

The rearrangement rules associated with the bond fragmentation are listed in Supplementary Table 9.

C3.12 Fig. 4: The CFM-ID 2.0 results are quite low which is inconsistent with the latest CASMI contest data (also blinded molecules). Also MAGMA results are surprisingly high, which is inconsistent with the latest CASMI contests. Here maybe false parameter settings are responsible, all the input and output parameters for each of the benchmark sets need to be shared. That also includes input/output results and all raw tables. The ChemDistiller paper (ChemDistiller: an engine for metabolite annotation in mass spectrometry; Bioinformatics, 34(12), 2018, 2096–2102) which is one of the later and accurately performing algorithms also comes to the conclusion that MetFrag, CSI-FingerID and CFM-ID accuracy ranks between (76-88% in the TOP5). So I believe something is wrong with the reported data here in this report, at least for CFM-ID.

R3.12 We thank the reviewer for capturing this bug. Our initial run of CFM-ID 2.0 did not include filtering by precursor m/z, which negatively impacted its accuracy scores. We added our own database filtering as a preprocessing step for CFM-ID 2.0 with a precursor tolerance of 0.02 Da. Previously we had included results for the Jaccard scoring method.

Moreover, all input datasets and databases have been listed in the “Datasets” subsection and supplementary material. For the output and raw results, we have added a new section “Data Availability” as follows:

Data Availability

All the benchmarking results and GNPS datasets search results are available online at <https://github.com/mohimanilab/molDiscovery>

We have also added a Supplemental Note S3 with parameter settings for all compared methods, replicated below.

Supplementary Note 3. Parameter settings for benchmarking.

Running CFM-ID. We used CFM-ID 2.0 associated with R31 at <https://sourceforge.net/p/cfm-id/code/HEAD/tree/>. Preprocessing consisted of converting compounds to predicted spectra using the cfm-predict binary. Following directions on the CFM-ID wiki (<https://sourceforge.net/p/cfm-id/wiki/Home/#cfm-predict>) we ran

```
cfm-predict SMILES 0.001 param_output0.log param_config.txt OUT_DIR 1 1
```

All parameters here are the defaults except that the `supress_exceptions` option was turned on. As suggested by the FAQ (<https://sourceforge.net/p/cfm-id/wiki/Home/#frequently-asked-questions>) we used the `metab_se_cfm` parameters. Scoring was done with the `cfm-id-precomputed` binary. According to the same FAQ, we repeated our single-energy data for each of the energy levels. We outputted 10 candidates using a relative tolerance of 10ppm and an absolute tolerance of 0.01 Da using the Jaccard (suggested for ESI-MS/MS) scoring method. Since CFM-ID does not have in-built filtering of a chemical database by precursor m/z we have added our own filtering to create a custom database for each spectrum, considering all compounds with mass within 0.02 Da of the mass of a $[M+H]^+$ adduct for the current spectrum.

Running MAGMa+. We used MAGMa+ 1.0.1. To preprocess our chemical compound database we created an SQLite database containing a single table using the following SQL command:

```
CREATE TABLE molecules ( id TEXT PRIMARY KEY, mim INTEGER NOT NULL,
charge INTEGER NOT NULL, natoms INTEGER NOT NULL, molblock TEXT,
inchikey TEXT, molform TEXT, name TEXT, reference TEXT, logp INT );
```

To fully replicate MAGMa+ we then built the same index they do after inserting all compounds using the SQL command:

```
PRAGMA temp_store = 2;
CREATE INDEX idx_cover ON molecules ( charge, mim, natoms, reference,
molform, inchikey, name, molblock, logp );
```

MAGMa+ does not support the MGF format. Each MGF spectrum was converted to MAGMa+ internal mass tree format via the following process: (i) peak intensities were normalized to range from 0 to 100, (ii) the precursor m/z was set to intensity 100, and (iii) mass tree was constructed using format 'PRECURSOR: 100 (mz1: norm_intensity1, mz2: norm_intensity2, ...)'

Each individual spectrum in the mass tree format was read via the `read_ms_data` sub-command using positive ionization mode and a precursor absolute error tolerance of 0.02 Da and a maximum product ion error tolerance of 0.01 Da. Each spectrum was then annotated

with the annotate subcommand using the parameter minimum peak intensity set to 0.

Running CSI:FingerID. We used SIRIUS 4.5.3 with SIRIUS lib 4.4.8 and CSI:FingerID 1.4.8. Compounds were preprocessed into a custom database using the custom-db option with a file with one SMILES per line. Options were set by exporting a command-line version of an analysis set using the SIRIUS GUI, command replicated below with placeholders for inputs.

```
sirius \  
  -i $SINGLE_MGF_SPECTRUM \  
  -o $OUTPUT_DIR \  
  config \  
  --AlgorithmProfile qtof \  
  --IsotopeMs2Settings IGNORE \  
  --MS2MassDeviation.allowedMassDeviation "10.0ppm (0.01 Da)" \  
  --NumberOfCandidatesPerIon 1 \  
  --Timeout.secondsPerTree 1000 \  
  --NumberOfCandidates 10 \  
  --FormulaSettings.enforced HCNOPS \  
  --Timeout.secondsPerInstance 0 \  
  --AdductSettings.detectable "[[M+H]+" \  
  --StructureSearchDB $STRUCTURE_DB_NAME \  
  --AdductSettings.fallback "[[M+H]+" \  
  --FormulaResultThreshold true \  
  --RecomputeResults true \  
  formula \  
  structure
```

Running MetFrag. We used version 2.4.5 of the MetFragCommandLine jar from the MetFragRelaunched repository. We used a slightly modified version of the parameter file used for the MetFrag submission in CASMI 2016, reproduced below.

```
#  
# database parameters -> how to retrieve candidates  
#  
#  
MetFragDatabaseType = LocalPSV  
LocalDatabasePath = $PSV_FILE  
#  
#  
peak matching parameters  
#  
FragmentPeakMatchAbsoluteMassDeviation = 0.01
```

```

FragmentPeakMatchRelativeMassDeviation = 5
PrecursorIonMode = 1
IsPositiveIonMode = True
#
# scoring parameters
#
# taken from CASMI 2016 positive mode
NumberMaximumPeaksUsed = 40
FingerprintPeakAnnotationFile = $METFRAG_BASE/peak_annotations_pos.txt
FingerprintLossAnnotationFile = $METFRAG_BASE/loss_annotations_pos.txt
MetFragScoreTypes = FragmenterScore,
AutomatedPeakFingerprintAnnotationScore,
AutomatedLossFingerprintAnnotationScore
MetFragScoreWeights = 0.378258289605048,
0.487761785135587,
0.133979925259365

FingerprintType = CircularFingerPrinter
LossFingerprintAnnotationBetaValue = 5e-04
LossFingerprintAnnotationAlphaValue = 0.0025
PeakFingerprintAnnotationBetaValue = 0.0125
PeakFingerprintAnnotationAlphaValue = 1e-04
#
# output
# SDF, XLS, CSV, ExtendedXLS, ExtendedFragmentsXLS
#
MetFragCandidateWriter = CSV
#
# following parameteres can be kept as they are
#
MaximumTreeDepth = 2
MetFragPreProcessingCandidateFilter =
UnconnectedCompoundFilter,IsotopeFilter
MetFragPostProcessingCandidateFilter = InChIKeyFilter
NumberThreads = 1
# Adding for each spectrum:
# PeakListPath
# IonizedPrecursorMass
# DatabaseSearchRelativeMassDeviation
# SampleName
# ResultsPath

```

C3.13 Dataset section: Filtering via Splash (spectral bases) should not be done, but rather filtering based on structures. Using SMILES for filtering could be error-prone, Inchikeys should be used for that case.

R3.13 We have transitioned from filtering via canonical SMILES to filtering via the first 14 characters of InChIKeys. When filtering GNPS we found that multiple slightly different

compounds, as identified by InChIKey filtering, had exactly the same spectra (Supplementary Fig. 1). To avoid ambiguity in evaluation we simply kept one of each of these. We have updated the “Datasets” subsection about these changes. See R3.8.

Supplementary Figure 1: Examples of different structures corresponding to the identical spectra in the GNPS spectral library. **a** splash10-00ei-0119850000-7e4cfe9a3a48492095dd, **b** splash10-0ab9-9000000000-41fe0c437d174e817929, and **c** splash10-0udi-9000000000-dc3aeb701482eda1fe7.

C3.14 The used application dataset is MSV000079450 (pseudomonas, a gram-negative bacterium) is a rather minimal dataset if we consider hundreds and thousands of other organisms including those from mammalian or plant species. So here a few additional datasets such as human blood plasma or common plants (rice or Arabidopsis) would be useful to evaluate the tool.

Also many of the annotated compounds are peptides, so I wonder what happened to the 10,000 small core metabolites from core pathways that are commonly annotated in bacteria. This is a major weakness and could be related to the algorithm or the dataset.

R3.14 We added molDiscovery benchmarking on two additional datasets, namely MSV000084092 (human serum, 57K MS/MS scans) and MSV000086427 (38 plant species, 209K MS/MS scans). We have updated the “Datasets” subsection of the Results section as follows:

The Pacific Northwest National Laboratory (PNNL) lipid library, included as part of GNPS, was used for lipid identification benchmarking. We selected positive mode spectra with $[M+H]^+$ adducts for which the provided lipid names in the PNNL lipid library could be resolved by the

LIPID MAPS Structure Database (LMSD) text search tool. This left us with 15,917 spectra corresponding to 316 unique compounds.

We also selected multiple small molecule databases for different benchmarking tasks. DNP contained 77,057 non-redundant compounds from dictionary of natural product [13]. MoNA DB contained 10,124 compounds from MoNA. AllDB contained 719,958 compounds from DNP, UNPD [27], HMDB [28], LMSD [29], FooDB [30], NPAtlas [31], KEGG [32], DrugBank [33], StreptomeDB [34], GNPS spectral library molecules, MIBiG [35], PhenolDB [36] and Quorumpeps [37] and in-house databases. The KNApSAcK [38] database contains 49,584 compounds of plant origin. Bioactive-PubChem (~1.3 million compounds) contained all the compounds from the PubChem database that are less than 2,000 Da and have at least one reported bioactivity.

For mass spectra datasets, we selected 46 high-resolution GNPS spectral datasets (about 8 million spectra in total) with paired genomic/metagenomic data available [39] for large-scale spectral searching and secondary metabolites identification. Moreover, we selected spectral datasets from various environment, including MSV000084092 (~57,000 spectra from human serum), MSV0000864274 (~209,000 spectra from 38 plant species) and MSV000079450 (~400,000 spectra from *Pseudomonas* isolates).

We added the following subsection to the Results section, to describe these computational experiments.

Searching human and plant spectra. To demonstrate the ability of molDiscovery to identify plant and human metabolites, we benchmarked the tool on two recent GNPS datasets. Plant spectra from 38 species (MSV000086427) was searched against AllDB complemented with the KNApSAcK database, one of the largest specialized databases of plant metabolites. Note that AllDB also includes the UNPD comprising many compounds with plant origin²⁷. Human serum spectra (MSV000084092) was searched solely against AllDB as this database has already included all compounds from the Human Metabolome Database (HMDB), the largest metabolomics database for *H. sapiens* and the LMSD, which contains biologically relevant lipids in human serum.

Since both datasets are not annotated, it is not possible to validate molDiscovery identifications by comparing them to ground truth. Instead, we compared the number of identifications in KNApSAcK/UNPD (HMDB/LMSD) to the number of identifications in the rest of AllDB for the plant (human serum) dataset. Supplementary Fig. 13 shows the number matches from the expected database (KNApSAcK/UNPD for plant and HMDB/LMSD for human serum data) versus the number of matches among the rest of compounds in AllDB. The ratio of the identifications in the expected databases is proportionally higher than the ratio of size of the expected databases to the size of AllDB. This bias grows with the increasing cutoff score. Note that AllDB (719,958 compounds) is an order of magnitude larger than KNApSAcK (49,584), HMDB (41,919), and LMSD (40,358), and three times larger than UNPD (229,358). The bias towards compounds in the searches of plant (human serum) data indirectly demonstrates the correctness of molDiscovery results. Moreover, many hits in AllDB may also represent true positive matches since this combined database includes plant- and human-related metabolites from databases other than KNApSAcK, UNPD, HMDB and LMSD.

Supplementary Figure 13: MolDiscovery results on **a** plant and **b** human serum datasets. The number of molecule-spectrum matches at various score thresholds in the searches against **a** KNApSAcK, UNPD and the rest of AllDB, and **b** HMDB, LMSD and the rest of AllDB. For KNApSAcK, UNPD, HMDB, and LMSD both top 1 (solid lines) and top 3 (dashed lines) accuracy identifications are counted.

For the concern regarding peptide discovery, we believe that being able to discover peptides is an advantage of our method and not a disadvantage. See our response R3.16 about the discovery of small and non-peptidic molecules.

Regarding the concern about small core metabolites from core pathways being missed, note that MSV000079450, and the majority of microbial datasets in the field of natural product discovery, are collected using a protocol that specifically captures exo-metabolites which are the metabolites secreted outside the microbial cell (e.g. secondary metabolites). We have updated the title of the subsection “MolDiscovery links small molecules to their biosynthetic gene clusters (BGCs).” of the Results section into

MolDiscovery links secondary metabolites to their biosynthetic gene clusters (BGCs)

C3.15 Lipids are important natural products, but they are completely ignored, here a lipid dataset could show the true performance of the tools

R3.15. We added a new experiment to evaluate the performance of molDiscovery by searching the Pacific Northwest National Laboratory lipid library against the LIPID MAPS Structure Database. See R3.14 for the updated datasets description. We also added a new subsection “Lipid Identification” to the Results section for this experiment:

Lipid Identification. We evaluated the performance of molDiscovery for lipid identification in searching spectra from the PNNL lipid library against 39,126 unique compounds from LMSD. We found that molDiscovery achieved a top 1 accuracy of 16.9%, a top 3 accuracy of 37.4%, a top 5 accuracy of 47.6%, and a top 10 accuracy of 63.0%. Although many lipids are extremely similar to one another (Supplementary Fig. 12), notably for 173 of the 316 unique compounds in the PNNL library, molDiscovery was able to find at least one spectrum where the top-scoring match corresponded to the correct compound. This experiment was run without any re-training of molDiscovery on lipid spectra.

Supplementary Figure 12: Structure of very similar lipids **a** LMGP01050034 and **b** LMGP01050035. Both compounds are candidates for spectrum CCMSLIB00003087761 from the PNNL lipids library. The correct compound **a** gets a score of -9.63 while the incorrect compound **b** gets a score of -7.42. Many of the lipid structures are very similar to each other, making it more difficult to distinguish them based on mass spectra

See also R1.7 for modifications to the text for an exploration of the ability of molDiscovery to distinguish very similar molecules.

Find also details on another descriptive experiment with the LMSD database and lipids in a human serum dataset in R3.14 above.

In addition, we also showed that molDiscovery ranked second in the identification of lipids in the GNPS spectral library benchmarking. See R3.24 for details.

C3.16 Most of the compounds annotated in the supplement are peptides. That is great, but then it would be important to say very early in the article abstract, results and discussions, that the algorithm mainly works for peptide annotations. That is fine, but that is exactly what the paper and supplement data show.

R3.16. We disagree with the reviewer and we argue that our method can also discover non-peptidic small molecules. For example, in Table 2 (now it is Table 3), among the 19 molecules identified by molDiscovery, 5 are not peptides, and 9 are less than 600 Da. We revise the subsection “MolDiscovery links secondary metabolites to their biosynthetic gene clusters (BGCs)” to highlight these discoveries. See R3.26.

We further revised the explanation of Fig. 7b (now it is Fig. 5b) to highlight the identification of non-peptidic small molecules from the GNPS spectral library (See R3.26)

In addition, we showed that molDiscovery was able to identify lipids by searching the PNNL lipid library against the LMSD (See R3.15).

Moreover, we add a table to the Supplementary Material to show the top identifications in each mass range (See R3.7, Supplementary Table 3).

C3.17 The paper focused on limited (peptidic) bond types. I want to know the expansibility of the method to some other bond types. Will this decrease the accuracy significantly? It's interesting to see the trends.

R3.17 In addition to peptidic (N-C) bonds that often break in peptides, molDiscovery also incorporates O-C and C-C bonds. During the revision, we also added S-C, O-P and P-C bonds to the list of bonds that molDiscovery breaks. We added a subsection "The effect of various bond types" to the Results section, to explore the sensitivity of molDiscovery to addition/removal of each of these bond types (please see R1.3 for details). As can be seen, removal of O-C, N-C and C-C bonds decreases the accuracy of molDiscovery. Addition of S-C, P-C and O-P does not improve the performance much.

C3.18 The existing methods "cuts" the molecules arbitrarily, what about the aromatic system? For example, how to divide the benzene ring or large resonant structures?

R3.18 We have added the following sentence to the subsection "Construct fragmentation graphs of small molecules" of the Methods section to clarify this.

Currently, we naively regard benzene ring and large resonant structures as alternate carbon-carbon single bonds (C-C) and double bonds (C=C). As molDiscovery can only cut at C-C, C-N and C-O bonds, it will only cut single C-C bonds in benzene rings.

In addition, we also updated the Discussion section to discuss the complexity of fragmentation of aromatic systems like benzene rings. See R1.3 for details.

C3.19 The authors should benchmark MolDiscovery in more scenarios, for example, searching spectra from CASMI / MoNA / Metlin / NIST against PubChem / ChEMBL / other big molecular databases. This benchmark is very important to convince people that molDiscovery can be used for searching general metabolites / small molecules.

R3.19. We benchmarked molDiscovery on a high-resolution subset of MoNA (~6.5K spectra) against a bioactive subset of PubChem (~1.3M compounds). See R3.14 for the updated dataset description. Since the search against such a large molecular database is very time-consuming for most of the competing tools, we decided to compare molDiscovery accuracy only with the fastest competing approach, MAGMa+ (CFM-ID preprocessing is infeasible for 1.3M compounds). We added the following subsection to the Results section to describe this computational experiment. The real execution time for molDiscovery and MAGMa+ as well as time estimates for the rest of the methods are provided in the new subsection and a new Supplementary Table 4.

Scalability of molDiscovery to large molecular databases. To demonstrate molDiscovery performance in high-throughput analysis of general metabolites, we benchmarked its ability to identify high-resolution MoNA spectra against a combination of 1.3 million bioactive-PubChem compounds and the ground truth compounds of MoNA spectra.

The search against a database with more than a million compounds is prohibitively time-consuming. The projected running times for all considered approaches on the MoNA spectra are

shown in Supplementary Table 4. Due to resource limitations, we analyzed this dataset only with molDiscovery and MAGMa+, the fastest and the most memory-efficient tools among the competitors. We ran both tools on a server with 20 CPUs Intel Xeon 2.50 GHz. The real execution times of molDiscovery and MAGMa+ were 38 minutes and 1,988 minutes respectively. These numbers are in line with the single-thread projections in Supplementary Table 4. Fig. 8 shows results of molDiscovery and MAGMa+ for top 1 and top 3 ranked identifications. Supplementary Fig. 16 shows the detailed results for multiple ranks and various score thresholds. MolDiscovery demonstrates better or the same accuracy as MAGMa+ while achieving the results significantly faster.

Method	Projected Running time (d-h:m:s) on MoNA
molDiscovery	Preprocessing - 7-3:08:04 Running - 11:25:40
CFM-ID	Preprocessing - 2522-21:42:37 Running - 23:26:15
Dereplicator+	4-9:25:53
MAGMa+	67-9:41:16
MetFrag	291-21:01:36
CSI:FingerID	693-12:19:03

Supplementary Table 4: Projected running times of all methods on searching the MoNA subset (~6,500 spectra) against bioactive-PubChem (~1.3M compounds). Projections assume that all the methods scale with $O(mn)$ for search and with $O(m)$ for preprocessing where m is the number of compounds in the chemical database and n is the number of spectra being searched. Projections are computed based on the running times recorded when searching GNPS against DNP.

Figure 8: Accuracy of high-throughput identification of the MoNA mass spectra against bioactive-PubChem.

Supplementary Figure 16: a-b molDiscovery and c-d MAGMa+ accuracy in the search of the MoNA subset (~6,500 spectra) against bioactive-PubChem (~1.3M compounds). a,c specificity (% correct identifications) and the b,d number of correct identifications for top 1, 3, 5 and 10 ranked identifications are shown at different score threshold levels.

Minor Comments

C3.20 Abstract: “This is a challenging task as currently it is not clear how small molecules are fragmented in mass spectrometry.”

I think rephrasing that to “complicated” or difficult would be better fitted.

R3.20 We have replaced “challenging” with “complicated” in the Abstract.

C3.21 Abstract: “Recently, spectral libraries with tens of thousands of labelled mass spectra of small molecules have emerged”

Please rephrase. “labelled” in context of analytical chemistry usually means isotopically labelled. If you refer to metadata-enriched or annotated spectra, please say so.

R3.21 We have replaced “labelled” with “annotated”.

C3.22 Discuss negative ionization mode mass spectra (negative mode)

R3.22 We have evaluated molDiscovery on 3,964 spectra of negative ionization mode against the unique DNP database. We changed the subsection “Performance on doubly-charged spectra” into “Performance on doubly-charged and negatively-charged spectra”, and added the following sentence:

Moreover, we tested molDiscovery on 3,964 negatively charged spectra of the GNPS spectral library against the unique DNP database. Notably, without retraining on negative spectra, molDiscovery reached 36% accuracy (Supplementary Fig. 9).

Supplementary Figure 9: Database search accuracies of molDiscovery on 3964 spectra of negative ionization mode against the unique DNP database. Here, the precursor mass tolerance is 0.02 and the product ion mass tolerance is 0.01. The top $K = 1, 3, 5, 10$ accuracies are 36%, 59%, 68% and 78%, respectively. Note that we did not retrain molDiscovery on negatively charged spectra.

We also updated the Discussion section regarding negatively charged spectra. See R1.9 for details.

C3.23 Fig. 6, doubly charged species are important for peptides, but are relatively unimportant for small molecule approaches. The low 10-15% accuracy shows that. I would not include that Fig..

R3.23 We have moved the Fig. to supplementary material.

C3.24 Fig. 7 is a very interesting Fig., unfortunately very hard to read due to the small bars, here horizontal bars would be better. The two outliers for “non-metal” (please correct) and hydrocarbons are probably due to the training set used. Also CFM-ID uses a different library for peptides (not shown in the graph).

R3.24 Since two low accuracy classes “non-metal” and “hydrocarbons” have 4 and 2 data points in the non-redundant GNPS spectral library (Supplementary Fig. 5), the performance of molDiscovery is very low. We now use InChIKey for preprocessing and the latest version of the GNPS spectral library for training and testing. We revised the subsection “Performance on different compound classes and mass ranges”:

Performance on different compound classes and mass ranges. To evaluate how well molDiscovery performs on different compound classes, we used ClassyFire [40] to annotate the non-redundant GNPS spectral library. No single tool outperformed all the other tools in all the com-

pound classes (Fig. 6a). MolDiscovery performs better than all the other tools in two out of seven superclasses. Moreover, we compared the database search accuracies of these tools on polar and non-polar metabolites. MolDiscovery is the best performing tool for searching polar metabolites across all the polarity measures, while for non-polar metabolites there is no tool that is consistently better than others (Supplementary Fig. 11).

Similarly, molDiscovery performs equally or better than the competing methods in four out of six mass ranges (Fig. 6b). It is worth noting that in low mass ranges (< 600 Da) the prediction accuracies of all the methods are lower than their performance in high mass ranges (> 600 Da), which is probably due to the fact that large molecules tend to generate more fragment ions.

Figure 6: Database search accuracy of molDiscovery, Dereplicator+, CSI:FingerID, MAGMa+, and CFM-ID on molecules of a different superclasses from ClassyFire⁴⁰ and **b** different mass ranges on the GNPS spectral library compounds excluding CSI:FingerID training data. See Supplementary Fig. 10 for the number of compounds in each class and mass range.

C3.25 Fig. 8 please spell out MA, OT, TA – no idea what that means. Also consistent large caps/small caps would be nice.

R3.25 They are the abbreviated names of the compounds. We have changed them into PK-1, PK-2 and PK-3 to indicate they are polyketides. The full names of these compounds are listed in the caption of the Table. See R3.6.

C3.26 Also most of the compounds are peptides or analogs and are important but do not represent the full natural product space.

R3.26 As we only reported the natural products (secondary metabolites) that correspond to known biosynthetic gene clusters (BGCs) in the MIBiG database (the biggest known BGC database), it is possible that this database is biased towards peptide natural products.

Among the 19 molecules in Table 2, 5 are not peptidic natural products and 9 are less than 600 Da, indicating the ability of molDiscovery to discover small and non-peptidic natural products.

We revised the last sentence of the subsection “MolDiscovery links secondary metabolites to their biosynthetic gene clusters (BGCs)” as follows.

MolDiscovery successfully identified 19 molecules of various categories in microbial isolates that contain their known BGCs (Table 2), including 5 non-peptidic molecules and 9 molecules less than 600 Da.

Moreover, in Fig. 6, we showed that molDiscovery performs equally well or better than other methods in 4 out of 6 mass ranges, and molDiscovery is among the best performing methods in different compound types, indicating molDiscovery’s capability of handling various compound classes in different mass ranges (See R3.24). We also added a new experiment to show that molDiscovery is able to discover lipids (See R3.15).

C3.27 In the supplement the LogRank is defined as “ 2^{i-1} most intense peaks”, what does this mean? Why are the peaks are exponential?

R3.27. We removed the sentence “The number of peaks with $\logRank = i$ is 2^{i-1} ” and revised the paragraph to the following.

For example, $\logRank = 1$ represents the most intense peak, $\logRank = 2$ represents the next 2 most intense peaks, and $\logRank = 3$ represents the next 4 most intense peaks. In general, the peaks with rank between 2^{i-1} to $2^i - 1$ will be assigned to $\logRank = i$.

We use this strategy to reduce the number of parameters in our model and avoid overfitting. We added the following sentence to the paragraph:

Note that using \logRank , the number of parameters of the model reduces from 64 per *bondType* to only 7 per *bondType*.

C3.28 It is also better to give a clear definition for “null spectrum”.

R3.28. We added the following sentence to “Preferential fragmentation patterns in mass spectrometry.” of the Results section:

Note that the \logRank distribution of fragments due to the 2-cut CC_CC is similar to the null model distribution, defined as the distribution of log rank of peaks in the entire spectral dataset. This implies that in practice the fragmentation of CC_CC 2-cut rarely happens.

Reviewers' Comments:

Reviewer #1:

Remarks to the Author:

I believe the authors have addressed my concerns appropriately.

Reviewer #2:

Remarks to the Author:

This is a re-review of a re-submitted article. I find the author's changes to prior review to be credible and responsive to critiques provided. One has to balanced applicability of the new algorithms to the limitations of a single paper. The number of use cases in metabolomics is as varied as the number of molecular structures (i.e., there is a lot). Therefore, discussions around polar vs. non-polar metabolites, support of varied bond types are scholarly and important, but did not change my already positive view of the impact offered by this work in the original submission. The speed of the now-validated tool itself is impressive. The authors are earnest in responding to varied comments of the 3 reviewers, and show the applicability of MolDiscovery beyond secondary metabolites and peptides, important sub-areas of 'metabolomics'. The paper has been reworked, and contains many aspects and features (e.g., negative ion data) - and thus i am supportive of this work moved into its publication stage.

Reviewer #3:

Remarks to the Author:

I want to thank the authors for the detailed responses, I understand this can be frustrating and includes a lot of work.

I also think that the 49 pages response are somewhat excessive, but needed. I believe the overall quality of the article increased very much.